# UBR-1 deficiency leads to ivermectin resistance in *Caenorhabditis elegans*

**Yi Li**[1†], **Long Gong**[1†], **Jing Wu**[1†], **Wesley Hung**[2], **Mei Zhen**[2], **Shangbang Gao**[1*]

[1]Key Laboratory of Molecular Biophysics of the Ministry of Education, College of Life Science and Technology, Huazhong University of Science and Technology, Wuhan, China; [2]Lunenfeld-Tanenbaum Research Institute, Mount Sinai Hospital, University of Toronto, Toronto, Canada

## eLife Assessment

This **important** study allows for a better understanding of anthelmintic drug resistance in nematodes. The authors provide a detailed analysis of the role of UBR-1 and its underlying mechanism in ivermectin resistance using **convincing** behavioural and genetic experiments with *C. elegans*. Although the authors have addressed the concerns of the reviewers, it would be prudent for the authors to disclose the Dyf phenotype in ubr-1 mutants. The authors should at the very least report the Dyf phenotype and the experiment on which they base the argument that the Dyf phenotype does not affect their conclusions.

**\*For correspondence:**
sgao@hust.edu.cn

[†]These authors contributed equally to this work

**Competing interest:** The authors declare that no competing interests exist.

**Abstract** Resistance to anthelmintics, particularly the macrocyclic lactone ivermectin (IVM), presents a substantial global challenge for parasite control. We found that the functional loss of an evolutionarily conserved E3 ubiquitin ligase, UBR-1, leads to IVM resistance in *Caenorhabditis elegans*. Multiple IVM-inhibiting activities, including viability, body size, pharyngeal pumping, and locomotion, were significantly ameliorated in various *ubr-1* mutants. Interestingly, exogenous application of glutamate induces IVM resistance in wild-type animals. The sensitivity of all IVM-affected phenotypes of *ubr-1* is restored by eliminating proteins associated with glutamate metabolism or signaling: GOT-1, a transaminase that converts aspartate to glutamate, and EAT-4, a vesicular glutamate transporter. We demonstrated that IVM-targeted GluCls (glutamate-gated chloride channels) are downregulated and that the IVM-mediated inhibition of serotonin-activated pharynx $Ca^{2+}$ activity is diminished in *ubr-1*. Additionally, enhancing glutamate uptake in *ubr-1* mutants through ceftriaxone completely restored their IVM sensitivity. Therefore, UBR-1 deficiency-mediated aberrant glutamate signaling leads to ivermectin resistance in *C. elegans*.

## Introduction

Anthelmintic resistance (AR) to broad-spectrum antiparasitic macrocyclic lactones (MLs), which include ivermectin (IVM), avermectin (AVM), and doramectin (DOM), has emerged as a critical global concern in veterinary parasites (*Wolstenholme and Kaplan, 2012*; *Fissiha and Kinde, 2021*; *Kaplan and Vidyashankar, 2012*). The rise of anthelmintic-resistant nematodes poses a pressing issue, impacting not only livestock production but also animal welfare and human health. Overcoming this challenge requires a comprehensive understanding of the molecular mechanisms driving nematode resistance, which, in turn, will facilitate the development of innovative and effective parasite prevention strategies.

The nematode *Caenorhabditis elegans* (*C. elegans*), owing to its sensitivity to a majority of anthelmintic drugs and conservation of functional genes, serves as a crucial model organism for identifying genes related to IVM resistance (*Geary and Thompson, 2001*; *Holden-Dye and Walker,*

*2014*; *Burns et al., 2015*; *Martin et al., 2021*). Mutations in three *C. elegans* glutamate-gated chloride channel (GluCl) *genes—avr-15, avr-14*, and *glc*-1—have collectively demonstrated strong resistance to IVM (*Dent et al., 1997*; *Ghosh et al., 2012*; *Wolstenholme and Rogers, 2005*; *Dent et al., 2000*). Exogenous expression and crystallization X-ray structural analysis support the premise that these GluCls act as the primary targets of IVM (*Cully et al., 1994*; *Hibbs and Gouaux, 2011*). Further identification of IVM resistance-related genes, including P-glycoprotein transporters (*pgp-1, pgp-3, pgp-6, pgp-9, pgp-13*, and *mrp-6*), cytochrome oxidases (*cyp-14, cyp-34/35*), and gluta-thione S-transferases (*gst-4, gst-10*). (*Lespine et al., 2012*; *Ardelli and Prichard, 2013*; *Gerhard et al., 2020*), has revealed their role in enhancing IVM excretion or metabolism. These detoxi-fication genes are thought to be regulated by the ubiquitous transcription factor NHR-8, which encodes a nuclear hormone receptor (*Ménez et al., 2019*). While these studies have revealed a polygenic mechanism underlying IVM resistance, increasing reports of unknown causes of IVM resis-tance continue to emerge (*Dube et al., 2023*; *Dube et al., 2022*; *Robert et al., 2021*), suggesting that previously unrecognized or additional mechanisms regulating GluCls expression may await for further investigation.

UBR1, an E3 ubiquitin ligase and a pivotal component of the ubiquitin–proteasome system, facili-tates protein degradation (*Bachmair et al., 1986*; *Gardner et al., 2005*; *Varshavsky, 2014*; *Hershko et al., 1983*; *Haglund and Dikic, 2005*). UBR1 is implicated in substrate-specific recognition and metabolism, primarily through the N-end rule (*Sriram et al., 2011*). In humans, loss-of-function muta-tions in UBR1 result in Johanson-Blizzard syndrome (JBS), a rare autosomal recessive disorder charac-terized by a spectrum of developmental and neurological symptoms (*Johanson and Blizzard, 1971*; *Daentl et al., 1979*). In the nematode *Caenorhabditis elegans*, a single UBR-1 homolog retains all major conserved functional domains (*Chitturi et al., 2018*). LIN-28 is the sole identified substrate of *C. elegans* UBR-1, acting as an RNA-binding pluripotency factor crucial for seam cell patterning (*Weaver et al., 2017*). However, the absence of LIN-28 neither replicates nor ameliorates the move-ment defects observed in *ubr-1* mutants-stiff body bending during backward movement (*Chitturi et al., 2018*).

Our recent research has consistently shown that UBR-1 regulates neural outputs through glutamate homeostasis mechanisms. Loss of UBR-1 leads to elevated glutamate levels in animals, disrupting coordinated motor patterns, likely due to a compensatory decrease in the expression of excitatory glutamate receptors (*Chitturi et al., 2018*). Furthermore, *ubr-1* mutations disturb inherent asymmetric neural activity by activating inhibitory GluCls (GLC-3 and GLC-2/4), which in turn impairs rhythmic defecation motor programs (*Li et al., 2023*). These observations suggest that aberrant glutamate metabolism resulting from UBR-1 deficiency has a widespread effect on glutamatergic signaling path-ways. This leads to the question of whether *ubr-1* mutants cause comprehensive changes in IVM-targeted GluCls, thereby contributing to IVM resistance.

To investigate this possibility, we assessed the IVM sensitivity of wild-type and *ubr-1* mutant animals. We found that, in contrast to wild-type N2 animals, loss-of-function *ubr-1* mutants present a range of IVM-resistant phenotypes. N2 animals exhibit high fatality rates, abnormal development and motor defects, such as pharyngeal pumping or locomotion disabilities, upon exposure to IVM. In contrast, *ubr-1* mutants were almost insensitive to IVM. The elimination of the glutamate synthetase GOT-1 and the glutamate transporter EAT-4 completely reversed the IVM sensitivity of *ubr-1* mutants, highlighting the critical role of the glutamatergic signaling pathway. Using translational reporters, we found that IVM resistance in *ubr-1* mutants is caused by the functional downregulation of IVM-targeted GluCls, including AVR-15, AVR-14, and GLC-1. These receptors are activated by glutamate to facilitate chloride ion influx into pharyngeal muscle cells, thereby inhibiting muscle contractions and suppressing food intake in *C. elegans*. This downregulation of GluCls is presumably triggered by the aberrantly elevated glutamate-induced compensatory decrease, as the application of exogenous glutamate in wild-type animals partially replicated the IVM resistance observed in *ubr-1* mutants. GluCls downregulation is functionally validated, as evidenced by the diminished IVM-mediated inhi-bition of serotonin-activated pharyngeal $Ca^{2+}$ activity in *ubr-1* mutants. Moreover, pharmacologically reducing glutamate levels via the use of ceftriaxone (*Lee et al., 2008*; *Rothstein et al., 2005*) success-fully restored IVM sensitivity in *ubr-1* mutants. Thus, our study presents multiple lines of evidence for a novel IVM resistance mechanism linked to the deficiency of the ubiquitin ligase UBR-1 via a glutamate metabolism pathway.

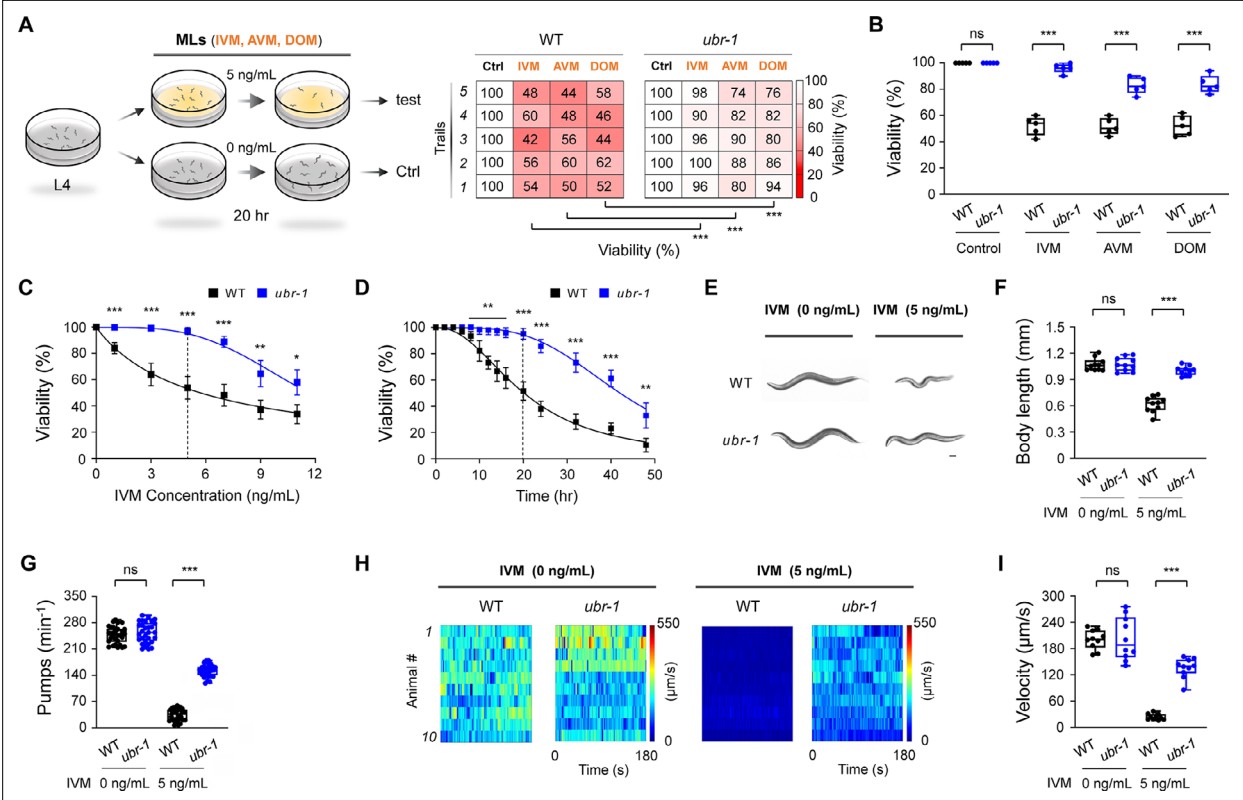

**Figure 1.** *ubr-1* exhibits IVM-resistant phenotypes. (**A**) *Left*: Schematic representation of the IVM resistance test in *C. elegans*. Well-fed L4 stage animals were transferred to plates containing OP50 bacteria with or without macrocyclic lactones (MLs) for 20 hr. *Right*: Representative grid plots illustrating the viability of wild-type and *ubr-1* animals in response to ivermectin (IVM, 5 ng/mL), avermectin (AVM, 5 ng/mL), and doramectin (DOM, 5 ng/mL). We used shades of red to represent worm viability on each experimental plate (n=50 animals per plate), with darker shades indicating lower survival rates. The viability test was repeated at least five times (five trials). (**B**) Quantification analysis of the viability of wild-type and *ubr-1(hp684)* mutants exposed to different MLs. *ubr-1(hp684)* mutants exhibit resistance to various MLs. (**C**) Dose–response curve depicting the viability of wild-type and *ubr-1(hp684)* mutants in the presence of varying concentrations of IVM. The IC$_{50}$ was 5.7 ng/mL for the wild type and 11.6 ng/mL for the *ubr-1* mutants. (**D**) Time-dependent effect of IVM exposure on animal viability. The IC$_{50}$ values were 20.2 hr for the wild type and 42.1 hr for the *ubr-1* mutants. (**E**) Representative images of worm size in the wild type and *ubr-1(hp684)* mutants with or without IVM treatment. Scale bar, 50 μm. (**F**) Quantitative analysis of body length in different genotypes with or without IVM. (**G**) Quantification of the average pharynx pump number in animals with or without IVM. (**H**) Raster plots illustrating the locomotion velocity of individual animals in the absence (0 ng/mL) and presence (5 ng/mL) of IVM. n=10 animals in each group. (**I**) Quantification of the average velocity in different genotypes with or without IVM treatment. ns, not significant, *p<0.05, **p<0.01, ***p<0.001 by Student's *t* test. The error bars represent the SEM.

The online version of this article includes the following figure supplement(s) for figure 1:

**Figure supplement 1.** Dose- and time-dependent patterns of IVM resistance observed in *ubr-1* mutants.

**Figure supplement 2.** IVM resistance across various *ubr-1* mutant alleles.

## Results

### Dose- and time-dependence of IVM resistance in *ubr-1* mutants

To further elucidate the mechanisms of AR, we employed a simplified assay derived from prior research (*Figure 1A*). We assessed the survival of nematodes on media infused with antiparasitic MLs. In essence, synchronized L4-stage worms were placed on NGM plates with OP50 bacteria and treated with various MLs, including IVM (5 ng/mL), AVM (5 ng/mL), or DOM (5 ng/mL). Twenty hours postexposure, we quantified the surviving worms and calculated the percentage of viability relative to the initial population.

Our observations revealed a significant decline in the viability of wild-type nematodes following ML exposure (IVM 52.0 ± 3.2%, AVM 51.6 ± 2.9%, DOM 52.4 ± 3.4%, in contrast to the vehicle control 100 ± 0.0%), confirming the susceptibility of *C. elegans* to MLs (*Figure 1B*). In contrast, *ubr-1(hp684; lf)* mutants presented remarkable resistance to ML treatments (IVM, *ubr-1* 96.0±1.7% vs vehicle

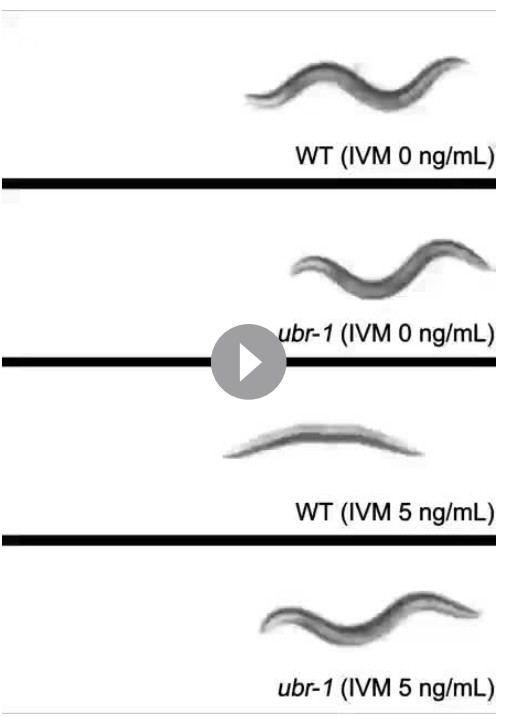

**Video 1.** Locomotion and body-length phenotypes of the wild type and *ubr-1* mutants before and after treatment with IVM. Representative free-moving worms of the wild type and *ubr-1(hp684)* mutants with or without IVM (50 ng/ml) treatment for 20 hr. The video shows the worm heading left. Scale bar, 50 µm.
https://elifesciences.org/articles/103718/figures#video1

control 100 ± 0.0%) when juxtaposed with their wild-type counterparts (IVM, WT 52.0 ± 3.2%; *Figure 1B*). This trend of increased survival was consistently observed with *ubr-1(hp684)* mutants treated with AVM (WT 51.6±2.9% vs *ubr-1* 82.8 ± 2.9%) and DOM (WT 52.4±3.4% vs *ubr-1* 83.6 ± 3.1%). These findings reveal the pronounced ML resistance characteristics of *ubr-1* loss-of-function mutants.

We further investigated the dose- and time-dependent resistance of *ubr-1(hp684)* to select IVM as a proxy for MLs because of its efficiency and common application. We tested a spectrum of IVM concentrations ranging from 1 ng/mL to 11 ng/mL (*Figure 1—figure supplement 1A*). The data revealed that *ubr-1(hp684)* mutants exhibited resistance at all tested doses (*Figure 1C*), resulting in a doubling of the half-maximal inhibitory concentration ($IC_{50}$) — from 5.7 ng/mL in wild-type N2 worms to 11.6 ng/mL in *ubr-1* mutants. Time-dependent IVM resistance was also compared between the wild type and *ubr-1* mutants (*Figure 1—figure supplement 1B*). The duration of half-maximal inhibition in the wild type was approximately 20.2 hr, which increased to 42.1 hr in the *ubr-1(hp684)* mutants (*Figure 1D*). To highlight the disparity in IVM resistance, we utilized an assay with 5 ng/mL IVM for a 20 hr period (indicated by the vertical dashed lines in *Figure 1C and D*) for subsequent experiments, unless otherwise specified.

## Additional IVM resistance phenotypes of *ubr-1*

In addition to viability, *ubr-1* mutants maintained normal growth despite IVM exposure. After 20 hr of IVM, the body size of the wild-type animals significantly decreased (*Geary, 2005*), resulting in a marked decrease in the middle line body length (control, 1.07±0.02 mm; IVM, 0.62±0.03 mm; *Figure 1E*). In contrast, the size of the *ubr-1* mutants (control, 1.08±0.02 mm; IVM, 1.00±0.02 mm) was largely preserved, indicating that the *ubr-1* mutants developed into typical young adults (*Figure 1F*). Notably, while IVM exposure nearly halted pharyngeal pumping in wild-type animals (control 247.4±4.1 min⁻¹, IVM 34.6±2.8 min⁻¹), *ubr-1* mutants sustained considerable pumping activity (control 254.1±5.0 min⁻¹, IVM 152.9±3.0 min⁻¹; *Figure 1G*). Motor functions were similarly resistant to IVM in *ubr-1* mutants (*Video 1*). The velocity of free movement under IVM treatment was significantly greater in *ubr-1* mutants (135.9±7.2 µm/s) than in wild-type animals (23.8±2.1 µm/s; *Figure 1H and I*, *Figure 1—figure supplement 1C*). No significant differences in viability, body length, pharyngeal pumping, or locomotion speed were detected between *ubr-1* and wild-type animals in the absence of IVM treatment.

These results collectively indicate that *ubr-1* loss-of-function mutants present a comprehensive spectrum of phenotypes associated with IVM resistance.

## Different *ubr-1* mutants exhibit consistent IVM resistance

The *ubr-1* gene is functionally conserved from yeast to humans. In nematodes or filarial worms, *ubr-1* also exhibited high sequence similarity (*Figure 1—figure supplement 2A*). The *hp684* allele contains a premature stop codon, resulting in the truncation of the final 194 amino acids of UBR-1 (Q1864X; *Figure 1—figure supplement 2B*). To validate functional congruence, we expanded our study to include additional *ubr-1* mutant alleles. These alleles, which are genetically predicted to lack one or

more critical conserved domains, including *hp865*, which replaces the entire RING finger domain with an SL2-NLS::GFP, and *hp821 hp833* (E34X, E1315X), each introducing stop codons that suggest the absence of all functional domains. All these alleles demonstrated IVM resistance comparable to that of the *hp684* allele. Specifically, the viability of *hp865* (81.6 ± 3.8%) and *hp821 hp833* (76.4 ± 2.6%) increased following IVM treatment (*Figure 1—figure supplement 2C*). Similarly, the body lengths of *hp865* (0.93±0.02 mm) and *hp821 hp833* (0.90±0.02 mm) were not diminished by IVM exposure (*Figure 1—figure supplement 2D*). Pharyngeal pumping and spontaneous movement velocities were also significantly greater in *hp865* (154.2±3.4 min⁻¹, 100.2±7.4 µm/s) and *hp821 hp833* (134.9±3.7 min⁻¹, 86.1±6.9 µm/s) than in the wild type (*Figure 1—figure supplement 2E and F*). These findings demonstrate that various *ubr-1* mutants exhibit a consistent IVM-resistant phenotype. While we have identified relevant sequences in parasitic nematodes including *Onchocerca volvulus*, *Brugia malayi*, and *Toxocara canis*, potential mutations in *ubr-1*-like genes in these parasitic nematodes may lead to IVM resistance.

## Glutamate induces IVM resistance in wild-type worms

Systemic elevation of glutamate levels in *ubr-1* mutants has been documented (*Chitturi et al., 2018*; *Kwon et al., 2001*; *Hwang et al., 2011*), prompting an investigation into whether this increase is the key determinant of the observed resistance to IVM.

To test this hypothesis, we pretreated wild-type N2 animals with glutamate to mimic the excessive glutamate environment in *ubr-1* mutants (*Figure 2A*). During these trials, wild-type larvae were exposed to L-glutamate (20 mM) on NGM plates from the L1 to L4 stages before their transfer to IVM-containing plates. Strikingly, these glutamate-pretreated animals presented a significant increase in IVM resistance (*Figure 2B*).

Specifically, the viability of these glutamate-exposed animals increased from 52±3.2%–77.2 ± 1.5% compared with that of their nonexposed counterparts (*Figure 2C*). A corresponding increase in body size was noted, from 0.62±0.03 mm to 0.76±0.01 mm (*Figure 2D*). Motor activity similarly improved, with velocities increasing from 23.79±2.1 µm/s to 62.4±7.6 µm/s (*Figure 2E and F*). These findings clearly demonstrate that the external application of glutamate can induce IVM resistance in wild-type N2 animals. Interestingly, in *ubr-1* mutants, pretreatment with glutamate did not further increase IVM resistance, implying potential saturation of the glutamate effect in *ubr-1* mutants (*Figure 2B–F*).

In addition to glutamate, other metabolically related molecules, such as aspartate (Asp), a precursor in glutamate synthesis, and γ-aminobutyric acid (GABA), a product of glutamate metabolism, were also found to be elevated in the *ubr-1* mutant (*Chitturi et al., 2018*; *Li et al., 2023*). However, the cocultivation of wild-type N2 worms with aspartate or GABA did not confer IVM resistance (*Figure 2A–F*), suggesting the glutamate-specific regulation of IVM resistance. This finding highlights the specificity and sufficiency of glutamate in conferring IVM resistance in a wild-type context.

## Enhancing glutamate uptake restores IVM sensitivity in *ubr-1*

Given the likelihood that elevated glutamate levels underpin IVM resistance in *ubr-1*, mitigating this glutamate surplus could reverse resistance and reinstate IVM sensitivity. To evaluate this premise, we employed ceftriaxone (Cef; *Figure 2G*), which is known for enhancing glutamate uptake by upregulating excitatory amino acid transporter-2 (EAAT2; *Lee et al., 2008*; *Rothstein et al., 2005*), with the aim of reducing glutamate levels.

The results were striking: the IVM resistance typically observed in *ubr-1* mutants was completely counteracted by Cef treatment. While Cef (50 µg/mL) had a negligible effect on wild-type N2 animals, it fully restored IVM sensitivity in *ubr-1* mutants (*Figure 2H*). Moreover, this restoration followed a dose-dependent pattern, with a half-effect concentration of 5.14 µg/mL (*Figure 2H*). Cef treatment resulted in a decrease in the viability of *ubr-1* mutants from 96.0±1.7%–62.8 ± 2.3% in the presence of IVM (*Figure 2I*). Additional IVM resistance phenotypes in *ubr-1* mutants, such as body size, pharyngeal pumping frequency, and locomotion velocity, were fully ameliorated by Cef (*Figure 2J–L*).

In line with its classification as a beta-lactam antibiotic, Cef effectively reduced the growth of the worm food source *E. coli* OP50, resulting in a reduced bacterial lawn on NGM plates (*Figure 2—figure supplement 1A and B*). However, simply diminishing the OP50 bacterial quantity on NGM plates without Cef did not restore IVM sensitivity in *ubr-1* mutants (*Figure 2—figure supplement 1C–F*), suggesting that food scarcity is not a critical factor. Similarly, increasing the number of OP50

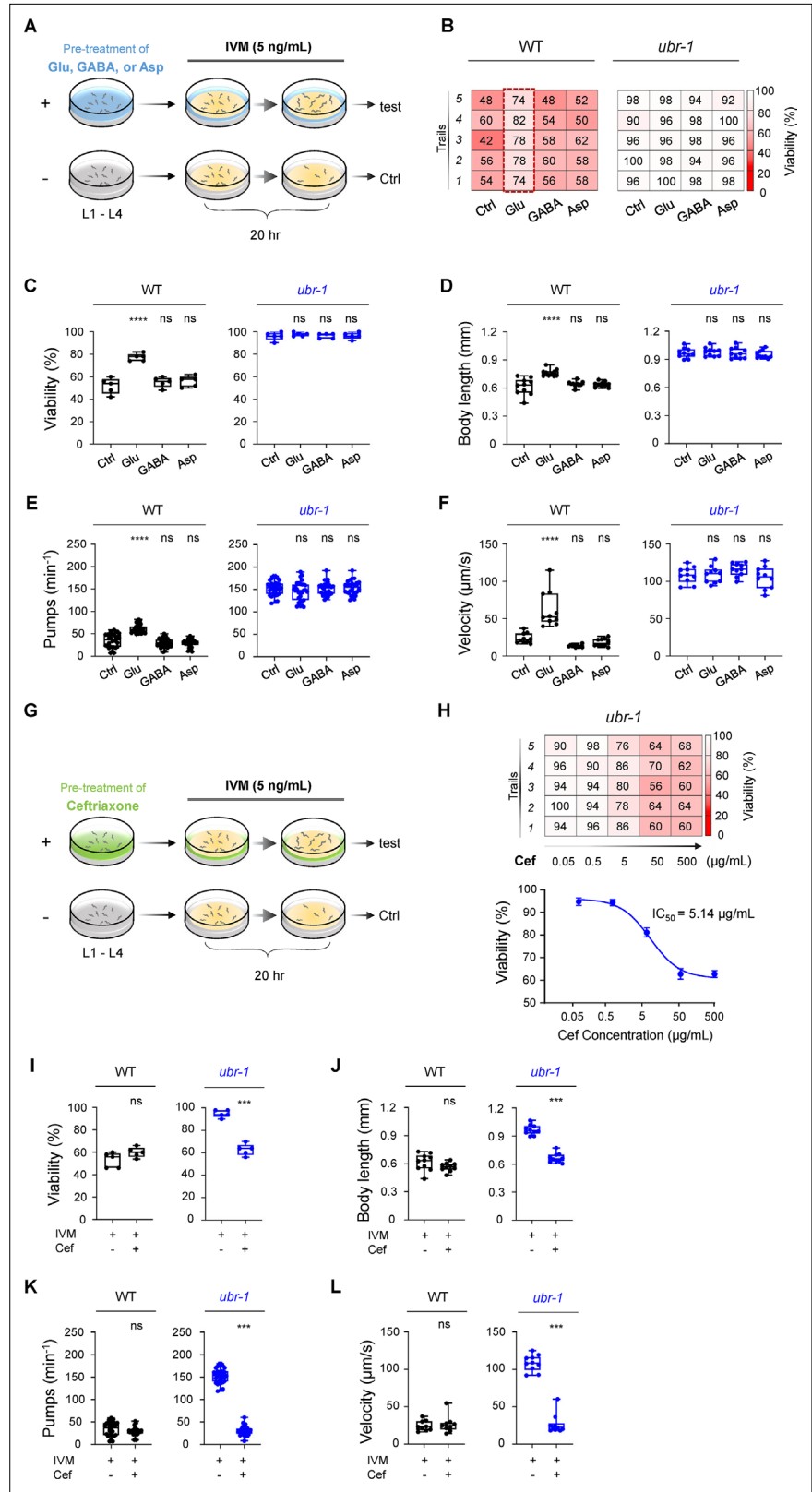

**Figure 2.** Glutamate mimics IVM resistance in wild-type N2 animals, and ceftriaxone fully reverses IVM resistance in *ubr-1* mutants. (**A**) Modified schematic representation of the pretreatment approach in the IVM resistance test in *C. elegans*. Synchronized L1 animals were cultured on plates containing OP50 bacteria and pretreated with glutamate (Glu, 5 mM), γ-aminobutyric acid (GABA, 5 mM), or aspartate (Asp, 5 mM) until they reached

*Figure 2 continued on next page*

*Figure 2 continued*

the L4 stage. Pretreated animals were then transferred to plates containing additional IVM (5 ng/mL) for 20 hr. (**B**) Representative grid plots illustrating the viability of wild-type and *ubr-1* animals pretreated with different glutamate metabolites. (**C–F**) Quantitative analysis of viability, body length, pharyngeal pump rate, and locomotion velocity in wild-type and *ubr-1* mutants following exposure to various glutamate metabolites. Wild-type worms treated with glutamate displayed resistance to IVM. However, glutamate did not affect the IVM resistance of the *ubr-1* mutant. ns, not significant, \*\*\*\*p<0.0001 by one-way ANOVA in C-F. (**G**) Diagram illustrating the ceftriaxone pretreatment procedure in the IVM resistance test. Synchronized L1 animals were cultured on OP50-fed plates with or without ceftriaxone (50 µg/mL) until the L4 stage. The animals were subsequently transferred to plates seeded with additional IVM (5 ng/mL) for 20 hr. (**H**) *Upper*: Representative grid plots depicting the viability of *ubr-1* animals at different concentrations of ceftriaxone. *Bottom*: Dose−response curve illustrating the amount of ceftriaxone required for viability in *ubr-1*(*hp684*) mutants (IC$_{50}$=5.1 µg/mL). (**I–L**) Quantification of viability, body length, pharyngeal pumping, and locomotion velocity in the wild type and *ubr-1* mutants in the absence and presence of ceftriaxone. Ceftriaxone fully restored the sensitivity of *ubr-1* mutants to IVM, comparable to the levels observed in wild-type N2 animals. ns, not significant, \*\*\*p<0.001 by Student's *t* test in I-L. Error bars represent SEM.

The online version of this article includes the following figure supplement(s) for figure 2:

**Figure supplement 1.** Ceftriaxone restores IVM sensitivity in *ubr-1* mutants through mechanisms beyond its antibacterial properties.

bacteria present on NGM plates containing Cef did not affect its ability to restore IVM sensitivity in *ubr-1* mutants (*Figure 2—figure supplement 1C–F*). These findings indicate that the antimicrobial action of ceftriaxone is not related to its ability to restore IVM sensitivity in *ubr-1* mutants.

Hence, these results indicate that pharmacologically reducing glutamate levels in *ubr-1* reestablishes IVM sensitivity, leading to the hypothesis that aberrant glutamate metabolism underlies IVM resistance in *ubr-1*.

## Restoration of *ubr-1* IVM sensitivity through glutamatergic signaling pathway regulation

Our previous research revealed the involvement of UBR-1 in regulating glutamate metabolism, which is essential for coordinated reversal bending and rhythmic defecation motor programs (*Chitturi et al., 2018*). The absence of UBR-1 also leads to perturbations in the balance between the GABAergic and glutamatergic signaling pathways (*Li et al., 2023*). In *C. elegans*, IVM sensitivity is marked by the sustained activation of glutamate-gated chloride channels (GluCls) by IVM (*Dent et al., 2000*). These glutamate-related findings led us to explore whether the resistance of *ubr-1* mutants to IVM could be attributed to alterations in the glutamate pathway.

At glutamatergic synapses, a series of enzymatic reactions link glutamate to neurotransmitter recycling. In *C. elegans*, glutamate-oxaloacetate transaminases, or GOT enzymes, facilitate amino group conversion between aspartate and α-ketoglutarate (α-KG) to oxaloacetate (OAA) and glutamate (*Figure 3A*; *Hayashi et al., 2003*; *Hirotsu et al., 2005*). We began our exploration by assessing IVM sensitivity following the disruption of GOT-1. While single got-1 loss-of-function mutants displayed IVM sensitivity similar to that of the wild type, ubr-1; got-1 double mutants presented a significant increase in IVM sensitivity, reverting to wild-type levels of viability, body length, and motor activity, in stark contrast to ubr-1 single mutants (*Figure 3B–E*). These findings indicate that impeding glutamate synthesis can effectively restore IVM sensitivity in *ubr-1* mutants.

Glutamate, a crucial excitatory neurotransmitter, is packed into synaptic vesicles by the vesicular glutamate transporter (VGLUT) and released into the synaptic cleft to activate postsynaptic receptors. We next explored the impact of inhibiting glutamate transport in *ubr-1* mutants by eliminating EAT-4, the sole VGLUT in *C. elegans* (*Lee et al., 1999*). Notably, the absence of EAT-4 in *ubr-1* mutants dramatically reversed their IVM resistance, as evidenced by increased viability (*ubr-1; eat-4* 66.8 ± 3.5%), reduced body length (*ubr-1; eat-4* 0.56±0.02 mm), increased pharyngeal pumping (*ubr-1; eat-4* 46.7±3.6 min$^{-1}$), and improved locomotion speed (*ubr-1; eat-4* 83.5±7.4 µm/s) in *ubr-1; eat-4* double mutants (*Figure 3B–E*).

In summary, these results underscore the importance of glutamate synthesis and synaptic transport in restoring IVM sensitivity in *ubr-1* mutants, suggesting a role for synaptic glutamate signaling in the resistance of *ubr-1* to IVM. The IVM resistance phenotypes of wild-type N2 animals were not rescued

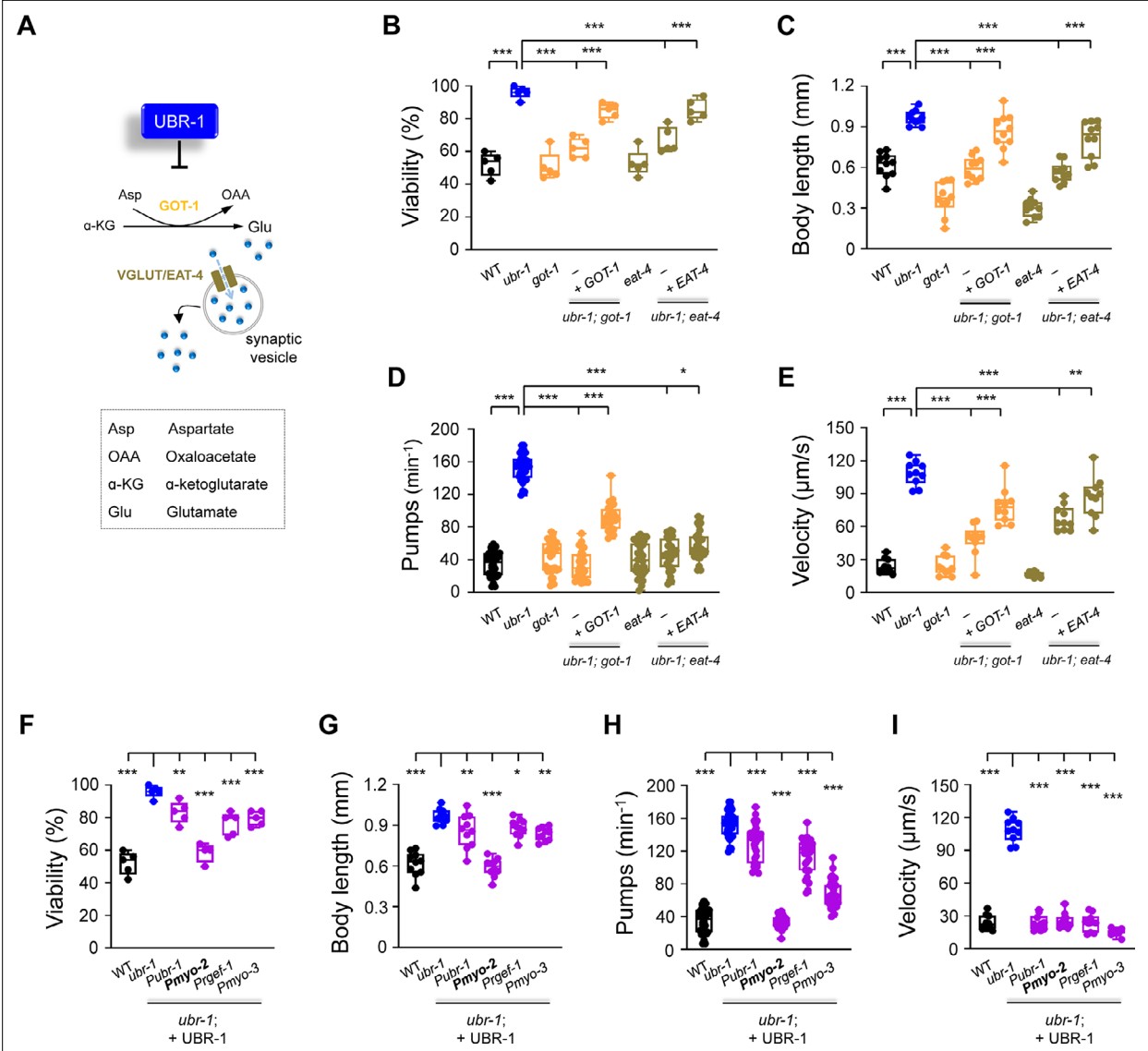

**Figure 3.** Deletion of *got-1* and *eat-4* suppresses *ubr-1*'s IVM resistance. (**A**) Schematic of the pathway for glutamate synthesis through transaminase GOT-1 and loading via vesicular transport VGLUT/EAT-4. (**B–E**) Quantification of viability, body length, pharyngeal pump rate, and locomotion velocity in different genotypes. The removal of *got-1* and *eat-4* significantly restored sensitivity to IVM (5 ng/mL) in *ubr-1* mutants, whereas the re-expression of GOT-1 (+GOT-1) and EAT-4 (+EAT-4) partially reinstated IVM resistance in the respective double mutants. (**F–I**) IVM resistance phenotypes were rescued by expression of UBR-1 (+UBR-1) driven by its own promoter (P*ubr-1*) or the pharynx muscle-specific promoter (P*myo-2*). For the viability test in F, n=50 animals per plate and repeated at least five times (five trials), n≥10 animals in G-I, *p<0.05, **p<0.01, ***p<0.001 by one-way ANOVA. All statistical analyses were performed against *ubr-1* mutant. Error bars, SEM.

The online version of this article includes the following figure supplement(s) for figure 3:

**Figure supplement 1.** Knockdown of UBR-1 induces IVM resistance phenotypes.

by ceftriaxone (*Figure 2I–L*), suggesting that ceftriaxone appears to be effective only against excess glutamate. This discovery is particularly intriguing, as it reveals a previously unexplored role of glutamate metabolism in the regulation of IVM resistance.

## UBR-1 appears to regulate IVM resistance from the pharynx

UBR-1 is ubiquitously expressed across a variety of tissues, including the pharynx, body wall muscles, and neurons (*Figure 3—figure supplement 1A*; *Chitturi et al., 2018*; *Kwon et al., 2001*; *Hwang et al., 2011*). To pinpoint the critical tissue for the role of UBR-1 in IVM resistance, we conducted tissue-specific rescue assays via a functional plasmid. Our findings revealed that restoring UBR-1

expression under its endogenous promoter attenuated the IVM resistance observed in *ubr-1* mutants. Notably, when UBR-1 expression was driven by an exogenous pharyngeal muscle-specific promoter, it fully corrected the IVM resistance phenotypes of the *ubr-1* mutants (*Figure 3F–I*). These results imply that although UBR-1 is important in multiple tissues, its role in the pharynx is particularly critical in modulating IVM sensitivity.

To verify the tissue-specific function of UBR-1, we utilized RNA interference (RNAi) to selectively decrease *ubr-1* expression in the pharynx. This precise RNAi application to the pharynx indeed induced IVM resistance in wild-type N2 worms (*Figure 3—figure supplement 1B–E*). RNAi-mediated knockdown of *ubr-1* in nonpharyngeal tissues, such as neurons and body wall muscles, also elicits an IVM resistance profile. Our research revealed that animals in which *ubr-1* was knocked down specifically in the pharynx presented more pronounced IVM resistance phenotypes than did those in which *ubr-1* was knocked down in other tissues. This phenotype approached the level observed when *ubr-1* was knocked down via its native promoter, underscoring the unique contribution of the pharynx to *ubr-1*-mediated IVM resistance.

Taken together, these results confirm that UBR-1 predominantly regulates IVM resistance through its action in the pharynx.

### Downregulation of IVM-targeted GluCls in *ubr-1*

What mechanisms underlie the induction of IVM resistance by abnormal glutamate metabolism? Previously, we established that *ubr-1* mutants exhibit impaired GLR-1 glutamate receptor function in AVA/AVE interneurons (*Chitturi et al., 2018*). Three GluCls (*avr-15*, *avr-14*, *glc-1*) are recognized as pivotal

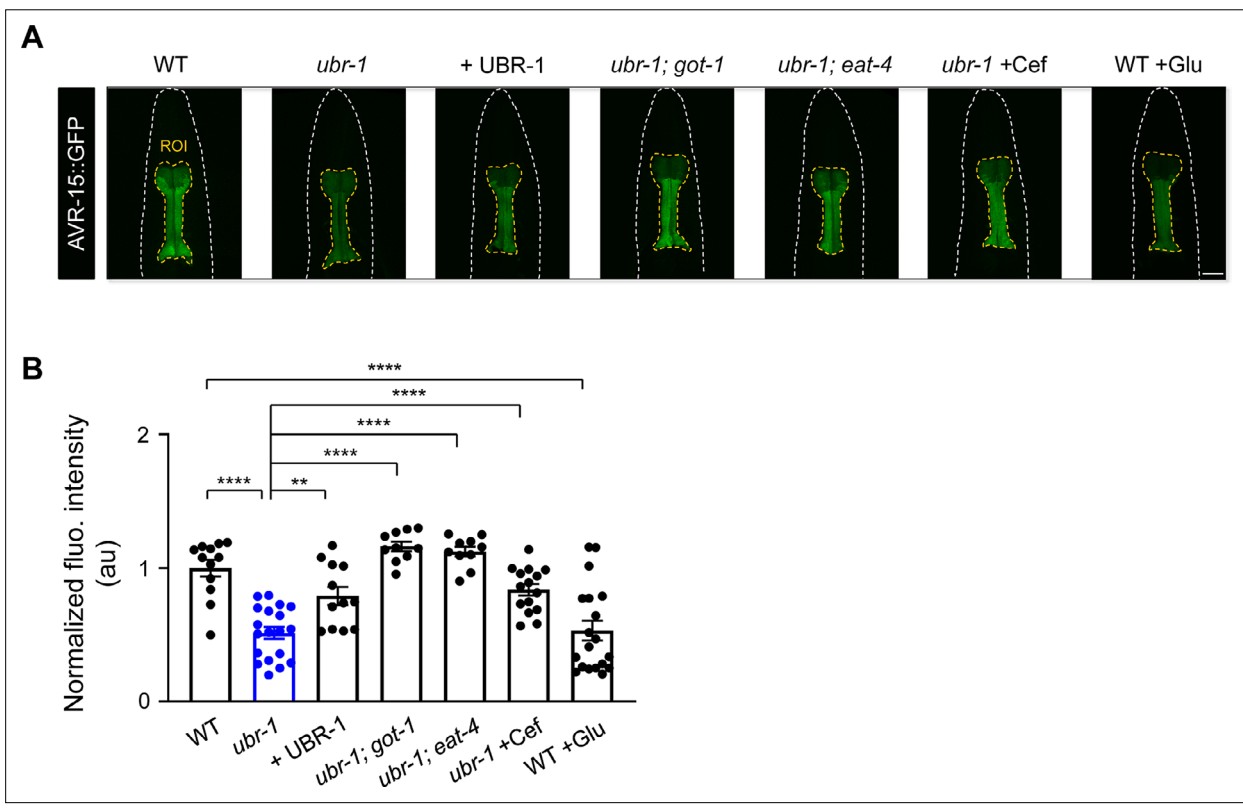

**Figure 4.** Downregulation of AVR-15 in *ubr-1* mutants. (**A**) Representative fluorescence intensity of AVR-15 in different genotypes. (**B**) Quantitative analysis of normalized fluorescence intensity revealed a reduction in AVR-15::GFP expression in *ubr-1* mutants. The diminished AVR-15 levels were rescued by overexpression of UBR-1 and genetically suppressed in *got-1* and *eat-4* mutants. Pharmacological intervention with ceftriaxone (50 µg/mL) successfully reinstated the pharyngeal expression of AVR-15::GFP. Conversely, glutamate administration reduced AVR-15 expression in the wild-type N2 strain. ROI, region of interest; n≥10 animals in each group. **p<0.01, ****p<0.0001 by one-way ANOVA. Error bars, SEM.

The online version of this article includes the following figure supplement(s) for figure 4:

**Figure supplement 1.** Downregulation of IVM-targeted GluCls in *ubr-1* mutants.

**Figure supplement 2.** Localization and intensity of AChRs and GABAARs in *ubr-1* mutants.

for IVM resistance in *C. elegans* (*Dent et al., 2000*). We sought to determine whether compromised IVM-targeted GluCls contribute to IVM resistance in *ubr-1* mutants.

To test this hypothesis, we created three transgenic strains with translational reporters for each GluCl under the control of their native promoters (P*avr-15*::AVR-15::GFP, P*avr-14*::AVR-14::GFP, P*glc-1*::GLC-1::GFP). Notably, AVR-15::GFP expression was pronounced in the pharynx, whereas GLC-1::GFP displayed a distinct head expression pattern, particularly concentrated in the head body wall muscles (*Figure 4A*, *Figure 4—figure supplement 1A*). AVR-14::GFP expression was localized primarily to various head and body neurons (*Figure 4—figure supplement 1A*). Intriguingly, compared with that in wild-type animals, the fluorescence intensity of these GluCls was markedly lower in *ubr-1* mutants (*Figure 4—figure supplement 1A and B*). The modulation of GluCl expression by *ubr-1* seems to be posttranscriptional, given that the RNA levels of the GluCl-encoding genes (*avr-15*, *avr-14*, *glc-1*) did not significantly differ (*Figure 4—figure supplement 1C*).

Crucially, the reduction in pharyngeal AVR-15::GFP fluorescence observed in *ubr-1* mutants was reversed upon reintroduction of UBR-1 (*Figure 4A*). Additionally, eliminating *got-1* and *eat-4* in these mutants mitigated the decrease in AVR-15::GFP expression within the pharynx (*Figure 4A and B*). Overexpression of AVR-15 also partially restored the sensitivity of ubr-1 mutant to IVM (*Figure 4—figure supplement 1D*). These results suggest a direct role of glutamate signaling in regulating GluCl expression levels. Notably, ceftriaxone pretreatment effectively reversed the diminished pharyngeal AVR-15::GFP levels in *ubr-1* mutants (*Figure 4A and B*). In addition to glutamate pretreatment eliciting IVM resistance in wild-type animals, the application of glutamate markedly diminished the pharyngeal AVR-15::GFP fluorescence in these same wild-type worms (*Figure 4A and B*). Interestingly, the ability of ceftriaxone pretreatment to rescue viability in *ubr-1* mutants was completely lost in the *avr-14; avr-15; glc-1* triple mutants (*Figure 4—figure supplement 1E*), confirming that ceftriaxone restores IVM sensitivity by reducing glutamate levels, which in turn upregulates GluCl receptors.

To further investigate the potential regulatory influence of *ubr-1* on a broader spectrum of synaptic receptors, we selected two distinct categories of receptors that are not activated by glutamate. This reduction was not observed for the excitatory acetylcholine receptor (UNC-29/AChR) or inhibitory GABA receptor (UNC-49/GABA$_A$R; *Figure 4—figure supplement 2*), suggesting that *ubr-1* may selectively affect glutamate-gated receptors (*Chitturi et al., 2018*).

Collectively, these results demonstrate that aberrant glutamate metabolism in the *ubr-1* mutant leads to the downregulation of IVM-targeted GluCls, which likely contributes to diminished IVM responsiveness, culminating in the IVM resistance observed in *ubr-1* mutants.

## IVM completely suppresses serotonin-evoked pharynx activity in wild-type animals but not in *ubr-1*

IVM acts by engaging target GluCls, leading to the irreversible inhibition of pharyngeal pumping, disrupting conduction between nerve and muscle cells and eventually killing parasites (*Dent et al., 1997*; *Dent et al., 2000*). The intricate dynamics of pharyngeal pumping in *C. elegans* rely on the coordinated interaction between the pharyngeal muscles and the excitatory pacemaker neuron MC, which is activated by serotonin (*Niacaris and Avery, 2003*).

To further examine the functional impairments of IVM-targeted GluCls, we conducted a direct examination of the impact of IVM on pharyngeal muscle cell activity. We generated a transgenic line expressing a Ca$^{2+}$ sensor, GCaMP6, in the pharynx, enabling real-time monitoring of pharyngeal muscle activity (*Figure 5A*; 'Materials and methods'). Using 5-HT to stimulate excitatory MC neurons, we induced intrinsic Ca$^{2+}$ activity within the pharyngeal muscles. In these assays, 5-HT triggered strong Ca$^{2+}$ transients across the pharyngeal muscle cells, from the terminal bulb through the isthmus to the anterior bulb (*Figure 5A*). To align with our quantified pumping behavior analysis, we focused our observations on the Ca$^{2+}$ intensity within the terminal bulb. We noted a dose-dependent response of pharyngeal muscle cells to 5-HT, with an estimated EC$_{50}$ of approximately 20.1 mM (*Figure 5B and C*). This assay provides a practical method to evaluate the functional impairment of IVM-targeted GluCls in *ubr-1*.

Interestingly, we observed that 5-HT induced similar Ca$^{2+}$ transients in both wild-type and *ubr-1* mutant worms. Furthermore, upon application of IVM (3 µM), 5-HT-elicited Ca$^{2+}$ activity was profoundly suppressed in the wild-type pharynx (*Figure 5D*), mirroring its effect on pharyngeal pumping. However, *ubr-1* mutants exhibited marked resistance to the suppressive effects of IVM, with only

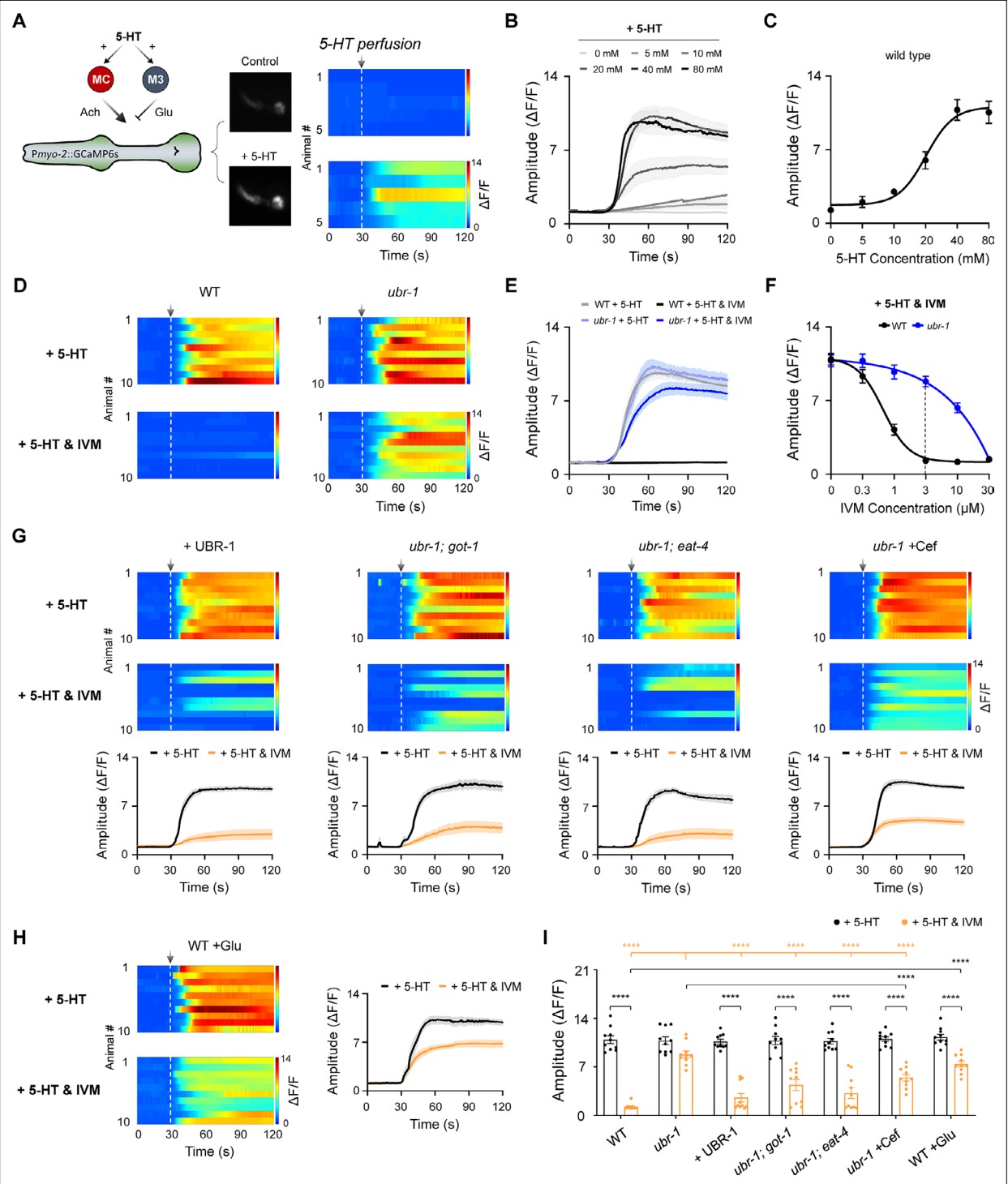

**Figure 5.** Reduced inhibition of serotonin-stimulated pharyngeal activity by IVM in *ubr-1* mutants. (**A**) *Left*: Schematic diagram of the pharynx motor circuit composed of an excitatory cholinergic motor neuron MC, an inhibitory glutamatergic motor neuron M3 and a group of compact muscles. Both types of motor neurons could be activated by serotonin (5-HT). *Right*: Representative pharynx muscle $Ca^{2+}$ response and kymograph evoked by 5-HT (20 mM). The dashed lines aligned with the arrows denote the initiation of perfusion. (**B**) Average $Ca^{2+}$ traces evoked by different concentrations of 5-HT. (**C**) Dose−response curve of 5-HT for the pharynx muscle $Ca^{2+}$ response in wild-type animals ($EC_{50}=20.1$ mM). n≥5 animals in each group. (**D**) Baseline-subtracted kymograph of the 5-HT-evoked pharynx muscle $Ca^{2+}$ response in wild-type and *ubr-1* mutants. IVM (3 µM) completely inhibited the wild-type pharynx $Ca^{2+}$ response, which was significantly reduced in *ubr-1* mutants. (**E**) Average $Ca^{2+}$ traces evoked by 5-HT in different genotypes with or without IVM. (**F**) The dose-dependent inhibitory effect of IVM on pharynx $Ca^{2+}$ was attenuated in *ubr-1* mutants (blue line) compared with wild-type animals (black line) ($IC_{50}=0.5$ µM in WT, while $IC_{50}=10.1$ µM in *ubr-1*). (**G**) The decreased IVM inhibition of serotonin-evoked pharynx activity in *ubr-1*

*Figure 5 continued on next page*

*Figure 5 continued*

was rescued by the restoration of UBR-1 expression (+UBR-1), genetic removal of *got-1* and *eat-4*, and pretreatment with ceftriaxone (+Cef, 50 µg/mL). (**H**) IVM-mediated inhibition of serotonin-evoked pharynx activity in wild-type animals was blocked by glutamate feeding (WT +Glu, 20 mM). n=10 animals in each group. (**I**) Quantification of the 5-HT-evoked peak amplitude of the pharynx muscle $Ca^{2+}$ response in the different genotypes. Black stars: ****$p<0.0001$ was used to compare intragroup differences or pharmacological groups via Student's *t* test; apricot stars: ****$p<0.0001$ was used to compare the differences with *ubr-1* in the '+5 HT and IVM' group via one-way ANOVA. Error bars, SEM.

a modest decrease in the amplitude of $Ca^{2+}$ transients (*Figure 5E*). Essentially, the capacity of IVM to inhibit 5-HT-driven pharynx activity was significantly compromised in *ubr-1* mutants. Additionally, consistent with its behavioral impact, IVM dose-dependently suppressed 5-HT-elicited pharynx $Ca^{2+}$ activity, with an estimated IVM $IC_{50}$ of approximately 0.5 µM for wild-type and 10.1 µM for *ubr-1* mutants (*Figure 5F*). These findings reveal the functional deficits of IVM-targeted GluCls in *ubr-1*.

In line with our earlier behavioral findings, the restoration of UBR-1 expression successfully reversed the inhibition of 5-HT-induced pharynx $Ca^{2+}$ by IVM in *ubr-1* mutants (*Figure 5G*). This corrective effect was similarly observed in *ubr-1; got-1* and *ubr-1; eat-4* double mutants (*Figure 5G and I*), reinforcing the association between decreased GluCl expression and attenuated IVM inhibition in *ubr-1* mutants. These results suggest a mechanism whereby elevated glutamate levels in *ubr-1* mutants may lead to glutamate-induced downregulation of GluCls, ultimately leading to IVM resistance.

To substantiate this proposed mechanism, we evaluated pharyngeal $Ca^{2+}$ activity in *ubr-1* mutants pretreated with Cef and in wild-type animals pretreated with glutamate. Consistent with our behavioral data, Cef treatment effectively reversed the inhibitory effect of IVM on 5-HT-evoked $Ca^{2+}$ activity in the pharynx of *ubr-1* mutants (*Figure 5G and I*), whereas glutamate exposure resulted in significant IVM resistance in wild-type animals (*Figure 5H–I*). This alignment between behavioral outcomes and $Ca^{2+}$ activity measurements further supports the notion that glutamate accumulation is a critical factor in the downregulation of GluCls and the ensuing development of IVM resistance in *ubr-1* mutants.

## Discussion

In this study, we identified a crucial role of the E3 ubiquitin ligase UBR-1 in *C. elegans*, which contributes to IVM resistance. Given the importance of UBR-1 in regulating glutamate metabolism, our research introduces a novel mechanism for IVM resistance that could be applicable to other ML AR scenarios. Notably, this glutamate-based mechanism operates upstream, affecting the expression levels of GluCls, a departure from the previously documented variations in detoxification transporters that facilitate drug efflux. Through detailed investigation, we demonstrated that UBR-1 is instrumental in maintaining physiological glutamate homeostasis, thereby governing the abundance of various glutamate receptors. In *ubr-1* mutants, perturbations in glutamate metabolism lead to reduced expression of primary IVM-sensitive GluCls, leading to decreased IVM-induced inhibition of serotonin (5-HT)-evoked pharynx $Ca^{2+}$ activity and subsequent resistance to IVM.

### A working model for UBR-1-mediated IVM resistance in *C. elegans*

Building upon our findings, we propose a working model that illustrates the role of the E3 ubiquitin ligase UBR-1 in mediating IVM resistance (*Figure 6*). In *C. elegans*, UBR-1 ensures functional glutamate homeostasis by inhibiting the activity of transaminase GOT-1 through a pathway that has yet to be resolved. A deficiency in UBR-1 leads to uncontrolled production of excess glutamate by GOT-1, which is then packed into synaptic vesicles for release via the vesicular glutamate transporter VGLUT/EAT-4. This excess glutamate results in the downregulation of glutamate receptors, including the IVM-targeted GluCls AVR-15, AVR-14, and GLC-1. The consequent reduction in these receptors diminishes the inhibitory effect of IVM on 5-HT-evoked pharynx $Ca^{2+}$ activity, highlighting the notable IVM resistance in *ubr-1* mutants. Furthermore, the exogenous application of glutamate induces IVM resistance in wild-type animals, whereas the pharmacological enhancement of glutamate transporter activity via ceftriaxone mitigates resistance in *ubr-1* mutants. These observations support the conclusion that IVM resistance in *ubr-1* mutants is intricately linked with aberrant glutamate metabolism.

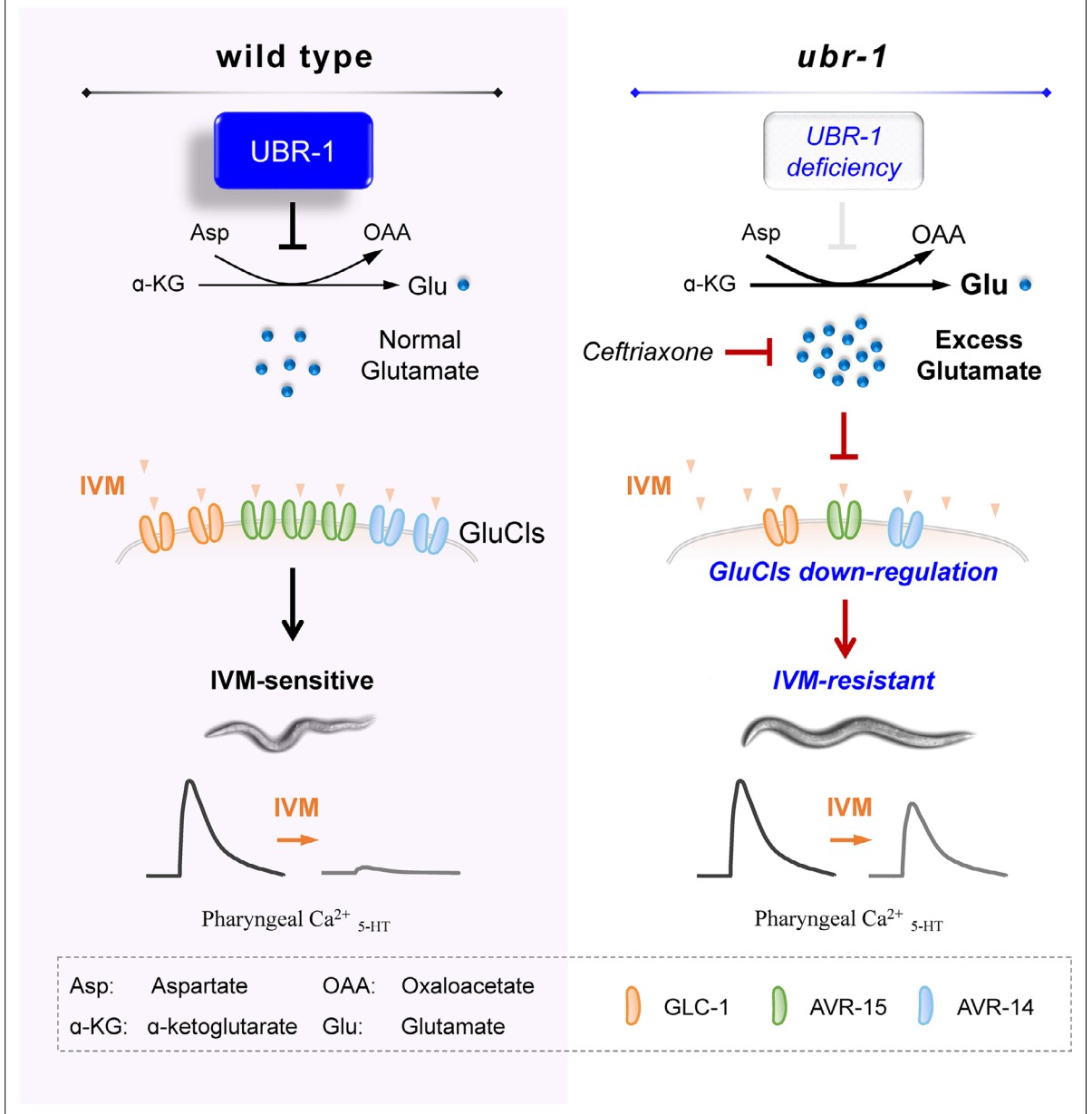

**Figure 6.** A working model. In wild-type animals, functional UBR-1 helps maintain balanced glutamate levels by inhibiting transaminase GOT-1 activity. In contrast, in *ubr-1* loss-of-function mutants, the absence of GOT-1 inhibition leads to excessive glutamate production. This glutamate excess, which can be reduced by ceftriaxone treatment, induces compensatory downregulation of IVM-targeted GluCls. This downregulation results in a diminished inhibitory response to IVM in the pharyngeal region, ultimately causing IVM resistance in *ubr-1* mutants.

## Unveiling a novel mechanism of anthelmintic resistance: aberrant glutamate metabolism

Glutamate is a central metabolic hub that connects glucose and amino acid metabolism in neurons and astrocytes. Disruptions in glutamate metabolism can precipitate an overabundance of glutamate, which in turn may trigger activity-dependent modulation of glutamate receptors. This phenomenon has been observed in *Drosophila*, where an increase in presynaptic glutamate concentrations leads to a notable reduction in both the number of glutamate receptors and the size of the synaptic field (*Featherstone et al., 2002*). Similarly, glutamate treatments have been shown to cause a loss of post-synaptic glutamate receptors (*Augustin et al., 2007*). In mammalian systems, prolonged exposure to glutamate has been demonstrated to specifically reduce RNA editing levels of AMPA receptors in

primary cortical neurons (*Bonini et al., 2015*). These findings suggest that a compensatory downregulation mechanism may be responsible for the observed decrease in glutamate receptors in response to excess glutamate.

In our study, we revealed that an overproduction of glutamate resulting from *ubr-1* mutations leads to the downregulation of IVM-targeted GluCls, which is subsequently linked to IVM resistance in *C. elegans*. This contrasts with previously elucidated mechanisms of IVM resistance, such as alterations in IVM-receptor binding sites that diminish their affinity, null mutations in IVM-targeted receptors, the upregulation of cellular efflux pumps, and enhanced drug degradation (*Fissiha and Kinde, 2021*). Our research introduces a novel metabolic mechanism involving glutamate receptors. We demonstrated that the levels of IVM-targeted receptors are subject to modulation via the glutamate metabolic pathway. This finding highlights the potential for glutamate accumulation, whether through metabolic dysregulation or exogenous administration, to confer IVM resistance, thereby offering fresh perspectives on the mechanisms behind AR. Our findings therefore reveal a new UBR-1-mediated pathway via GluCls that connects glutamate metabolism with IVM resistance.

Given the high degree of conservation in glutamate pathways between *C. elegans* and parasitic nematodes, the implications of our findings are far-reaching. This highlights the importance of developing new pharmacological agents aimed at modulating glutamate metabolism for the control of parasitic nematodes. Moreover, altering environmental glutamate concentrations—whether through modifications in the physiological environment of nematodes or changes in the host's diet—may increase the effectiveness of traditional ML anthelmintics.

## Expanding parasitic control through UBR-1 and glutamate clearance

Although it remains to be seen what percent of IVM-resistant parasites in the wild have disrupted glutamate homeostasis, there is a need for genetic markers of IVM resistance in livestock parasites that can be used to better track resistance and to tailor drug treatment. The discovery of UBR-1 as a resistance gene in *C. elegans* will provide a candidate marker that can be followed up in parasitic control.

Pharmacologically, the recommendation for combination anthelmintic treatments to slow the emergence of AR is well established, as they cover a similar range of parasites but act via different mechanisms (*Fissiha and Kinde, 2021*; *Jackson and Coop, 2000*). Our research proposes an innovative approach for parasite prevention: the clearance of glutamate. The use of ceftriaxone, which has been shown to completely recover IVM sensitivity in *ubr-1* mutants, presents a viable option to mitigate AR prompted by excess glutamate. A combination treatment that includes ceftriaxone and IVM-like anthelmintics is advisable. Moreover, the concurrent use of ceftriaxone with other antiparasitic drugs may prove effective in treating AR parasites.

This dual-treatment strategy offers several benefits, such as the widespread availability of ceftriaxone, its established FDA approval since 1982 (*Li et al., 2012*; *Fair and Tor, 2014*; *Wermuth, 2006*), the relatively low dose needed to increase IVM sensitivity, and its cost-effectiveness, particularly in the developing world.

Although the dynamics of AR may vary among different hosts, our study contributes a fresh outlook by revealing a new mechanism of IVM resistance and advocating alternative preventative measures. We hope that these insights will attract the attention of global health authorities such as the World Health Organization and spark additional research in the realm of parasite control.

## Materials and methods

### *C. elegans* strains and transgenic lines

*C. elegans* strains were cultured at 22 °C on nematode growth medium (NGM) plates seeded with *E. coli* OP50 as a food source (*Brenner, 1974*). The wild-type Bristol N2 strain was used, and the *ubr-1(hp684)* strains were sourced from an EMS mutagenesis screen. Additional *ubr-1* alleles were generated via the CRISPR-Cas9 system (*Chitturi et al., 2018*). Other genetic mutants were obtained from the *Caenorhabditis Genetics Center* (CGC). Transgenic worms were generated through microinjection via standard protocols, with target DNA plasmids at ~50 ng/μL coinjected with the marker plasmid P*myo-2*::RFP at 5–10 ng/μL. *Supplementary file 1* shows a detailed list of the strains used in this study.

## Ivermectin sensitivity assay

IVM and AVM (product number: PHR1380/31372) were purchased from Sigma-Aldrich (St. Louis, MO, USA) and then diluted in dimethylsulfoxide (DMSO). DOM (product number: D127678) was purchased from Aladdin (Shanghai, China). A stock solution of 40 mM IVM was prepared in 1 mL of DMSO. Initial ML resistance assays utilized IVM, AVM, and DOM at 5 ng/mL. Standard NGM plates with OP50 and 0.03% DMSO were used, with varying IVM concentrations for tests. L4 hermaphrodites were synchronized and exposed to IVM for viability assays. After 20 hr, the body length, pharyngeal pump rate, and locomotion velocity of the survivors were analyzed via a stereomicroscope (Axio Zoom V16, Zeiss). Each experiment was independently replicated at least three times.

## Animal viability and motor analysis

Animal viability was assessed by calculating the relative residual ratio: the number of viable worms after 20 hr of IVM incubation divided by the initial worm count. Viability (%), body length (mm), and velocity (μm/s) were measured via a custom MATLAB script (*Chen et al., 2022*).

For movement analysis, IVM-exposed worms were transferred to fresh OP50 plates. For locomotion velocity analysis, a single hermaphrodite was transferred to a 60 mm imaging plate. One minute after the transfer, a three-minute video of the crawled animal was recorded on a modified stereomicroscope (Axio Zoom V16, Zeiss) with a digital camera (acA2500-60um, Basler). Postimaging analyses utilized an in-house written MATLAB script. The central line was used for tracking. Images for velocity analysis from each animal were divided into 33 body segments. The midpoint was used to calculate the body length and velocity between each frame. The pharyngeal pumping frequency was directly calculated manually via video as pumps per minute ($min^{-1}$). Behavioral assays (including pumping frequency) were conducted on standard culture plates with freely moving worms.

## Pharmacological pretreatment

For glutamate application, glutamate (product number: G1251, Sigma-Aldrich) was dissolved in NGM at final concentrations of 0.2 mM, 1 mM, 5 mM, 10 mM, 20 mM, and 40 mM. Wild-type and *ubr-1* animals were exposed to glutamate from the L1 to L4 stages and then assessed after 20 hr on glutamate and IVM plates. The same experimental procedure was used for γ-aminobutyric acid (5 mM, product number: A2129, Sigma-Aldrich) and aspartate (5 mM, product number: A6202, Macklin, Shanghai, China).

In parallel, *ubr-1* mutants were treated with ceftriaxone sodium (product number: TC6154, Biobomei, Hefei, China) at final concentrations of 0.05 μg/mL, 0.5 μg/mL, 5 μg/mL, 50 μg/mL, and 500 μg/mL. Viability and motor ability were evaluated after L4 worms were transferred to plates containing ceftriaxone and IVM (5 ng/mL) for 20 hr.

## Molecular biology and RNA interference

All expression and rescue plasmids were generated via the multisite gateway system (*Magnani et al., 2006*). Three entry clones (Slot1, Slot2, Slot3) corresponding to the promoter, target gene and fluorescent marker genes were recombined into pDEST R4-R3 Vector II via the LR reaction, and expression constructs were subsequently obtained. The expression patterns of *ubr-1*, *avr-15*, *avr-14*, and *glc-1* were examined via promoter elements of 1.6 kb, 2.7 kb, 3.0 kb, and 1.14 kb, respectively. DNA fragments were PCR-amplified from N2 genomic DNA.

Double-stranded RNA (dsRNA) interference technology was used to suppress gene expression in specific tissues. Essential to this process is the expression of dsRNA, which is critical for achieving targeted neuronal or tissue-specific RNAi in *C. elegans* (*Fire et al., 1998*). The plasmid for RNAi was obtained by constructing Slot2 via the BP reaction of a multisite gateway system. The exon-rich region is usually selected as the interfering region (500–700 bp), the sense strand is selected as the target gene for forward interfering sequences (RNAi, sense), and the antisense strand is selected as the target gene for reverse interfering sequences (RNAi, antisense).

Detailed plasmid and primer information is available in *Supplementary file 2* and *Supplementary file 3*.

## RNA sequencing analysis

For RNA-seq analysis, synchronized L4 worms were washed with bath solution at least three times and harvested via centrifugation. Total RNA was isolated via the freeze–thaw method, and different samples were provided to Novogene for RNA-seq analysis. Transcripts per million (TPM), a standard gene expression normalization method that adjusts the expression of a gene to the number of transcripts per million (*Zhao et al., 2021*), was used to indicate quantitative gene expression. The TPM value takes into account the length of the gene and the sequencing depth and converts the expression of a gene to the number of transcripts per million by dividing the count value of each gene by its length and normalizing appropriately.

## Fluorescence microscopy

L4 stage worms were immobilized with 2.5 mM levamisole (Sigma–Aldrich, USA) on thin layers of agarose for imaging with a laser scanning confocal microscope (Olympus FV3000, Japan) with a 60×oil objective (numerical aperture = 1.35). The intensities of AChR and GABAAR in strains EN208 *kr208* [P*unc-29*::UNC-29::tagRFP] and KP5931 *nuIs283* [P*myo-3*::UNC-49::GFP] were measured. ImageJ software was used to process the images and perform intensity analysis, and the results are reported as the normalized fluorescence intensity per individual and are shown as the mean ± SEM.

## In situ Ca$^{2+}$ imaging

The strain SGA616 [P*myo-2*::GCaMP6s::wCherry] was used to facilitate calcium imaging of the pharyngeal muscles. One-day-old adult hermaphrodites were glued and transferred to bath solution for imaging with a 60 x water objective (Nikon, Japan; numerical aperture = 1.0). In the experiment, the epidermis behind head of the worm was gently punctured with a glass electrode (1–3 MΩ) to relieve internal pressure, which helps stabilize the calcium imaging. Serotonin (5-HT, product number: S31021, Yuanye, Shanghai, China) and IVM stock solutions were diluted to the desired concentrations via the following bath solution (in mM): NaCl 150; KCl 5; CaCl$_2$ 5; MgCl$_2$ 1; glucose 10; sucrose 5; and HEPES 15, pH 7.3 with NaOH, ~330 mOsm. The control bath solution, 5-HT alone, or a combination of 5-HT and IVM were administered 30 s after initiating the recording via a gravity-driven perfusion system (INBIO MPS-3, China). Fluorescence images were acquired with an excitation wavelength LED at 470 nm and a digital sCMOS camera (Hamamatsu ORCA-Flash 4.0V2) under a wide-field microscope (Nikon LV-TV) at a rate of 10 frames per second for 2 min. Data were collected from HCImage (Hamamatsu) and analyzed via Image-Pro Plus 6.0 (Media Cybernetics, Inc, Rockville, MD, USA) and ImageJ (National Institutes of Health). The fluorescence intensity of the region of interest (ROI) was defined as $F$, and the background intensity near the ROI was defined as $F_0$. The true muscle calcium fluorescence signal was obtained by subtracting the background signal from the ROI. $\Delta F/F_0 = (F - F_0)/F_0$ was plotted over time as a fluorescence variation curve.

## Statistical analysis

GraphPad Prism 8 was used for statistical analyses and graphing. Dose–response residual ratio (DRR), IC$_{50}$, and EC$_{50}$ values were determined via a sigmoidal dose–response model. Unpaired Student's $t$ test was used to analyze data between two groups, and ordinary one-way or two-way ANOVA was performed for statistical analysis when multiple groups of data were compared. The p values are indicated as follows: ns, not significant, *$p<0.05$, **$p<0.01$, ***$p<0.001$, ****$p<0.0001$. The error bars represent the SEMs.

## Acknowledgements

We thank *Caenorhabditis Genetics Center* and Xia-jing Tong for strains; Ying Wang, Ya Wang, Weicheng Duan and Bo Xiong for their technical support; and Lijun Kang, Xun Huang, and Hong Zhang for their reagents and valuable discussions. This research was supported by the Major International (Regional) Joint Research Project (32020103007 to SG), the National Natural Science Foundation of China (32371189 to SG), and the National Key Research and Development Program of China (2022YFA1206000 to SG).

## Additional information

### Funding

| Funder | Grant reference number | Author |
|---|---|---|
| Major International Joint Research Programme | 32020103007 | Shangbang Gao |
| National Natural Science Foundation of China | 32371189 | Shangbang Gao |
| National Key Research and Development Program of China | 2022YFA1206000 | Shangbang Gao |

The funders had no role in study design, data collection and interpretation, or the decision to submit the work for publication.

### Author contributions

Yi Li, Data curation, Formal analysis, Validation, Investigation; Long Gong, Jing Wu, Data curation, Validation, Investigation, Methodology; Wesley Hung, Mei Zhen, Resources, Validation, Methodology, Writing – review and editing; Shangbang Gao, Conceptualization, Supervision, Funding acquisition, Validation, Investigation, Methodology, Writing – original draft, Project administration

### Author ORCIDs

Mei Zhen (ID) https://orcid.org/0000-0003-0086-9622
Shangbang Gao (ID) https://orcid.org/0000-0001-5431-4628

Reviewer #1 (Public review): https://doi.org/10.7554/eLife.103718.3.sa1
Reviewer #2 (Public review): https://doi.org/10.7554/eLife.103718.3.sa2
Reviewer #3 (Public review): https://doi.org/10.7554/eLife.103718.3.sa3
Author response https://doi.org/10.7554/eLife.103718.3.sa4

# Additional files

### Supplementary files

Supplementary file 1. Mutants and transgenic strains used in this study.

Supplementary file 2. A list of plasmids used in this study.

Supplementary file 3. Sequence of primers used in this study.

Source data 1. Raw data used for statistical analysis in each panel.

MDAR checklist

### Data availability

All data generated for this study are available in the Supplementary Information files and from the public Zenodo repository (DOI: https://doi.org/10.5281/zenodo.15070036).

The following dataset was generated:

| Author(s) | Year | Dataset title | Dataset URL | Database and Identifier |
|---|---|---|---|---|
| Gao et al. | 2025 | UBR-1 deficiency leads to ivermectin resistance in *C. elegans* | https://doi.org/10.5281/zenodo.15070036 | Zenodo, 10.5281/zenodo.15070036 |

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
