## [Editor Report · eLife Assessment]

This **important** study allows for a better understanding of anthelmintic drug resistance in nematodes. The authors provide a detailed analysis of the role of UBR-1 and its underlying mechanism in ivermectin resistance using **convincing** behavioural and genetic experiments with *C. elegans*. Although the authors have addressed the concerns of the reviewers, it would be prudent for the authors to disclose the Dyf phenotype in ubr-1 mutants. The authors should at the very least report the Dyf phenotype and the experiment on which they base the argument that the Dyf phenotype does not affect their conclusions.

---

## [Referee Report · Reviewer #1 (Public review)]

Summary:

The drug Ivermectin is used to effectively treat a variety of worm parasites in the world, however resistance to Ivermectin poses a rising challenge for this treatment strategy. In this study, the authors found that loss of the E3 ubiquitin ligase UBR-1 in the worm *C. elegans* results in resistance to Ivermectin. In particular, the authors found that ubr-1 mutants are resistant to the effects of Ivermectin on worm viability, body size, pharyngeal pumping and locomotion. The authors previously showed that loss of UBR-1 disrupts homeostasis of the amino acid and neurotransmitter glutamate resulting in increased levels of glutamate in C. elegans. Here, the authors found that the sensitivity of ubr-1 mutants to Ivermectin can be restored if glutamate levels are reduced using a variety of different methods. Conversely, treating worms with exogenous glutamate to increase glutamate levels also results in resistance to Ivermectin supporting the idea that increased glutamate promotes resistance to Ivermectin. The authors found that the primary known targets of Ivermectin, glutamate-gated chloride channels (GluCls), are downregulated in ubr-1 mutants providing a plausible mechanism for why ubr-1 mutants are resistant to Ivermectin. Although it is clear that loss of GluCls can lead to resistance to Ivermectin, this study suggests that one potential mechanism to decrease GluCl expression is via disruption of glutamate homeostasis that leads to increased glutamate. This study suggests that if parasitic worms become resistant to Ivermectin due to increased glutamate, their sensitivity to Ivermectin could be restored by reducing glutamate levels using drugs such as Ceftriaxone in a combination drug treatment strategy.

Strengths:

- The use of multiple independent assays (i.e., viability, body size, pharyngeal pumping, locomotion and serotonin-stimulated pharyngeal muscle activity) to monitor the effects of Ivermectin

- The use of multiple independent approaches (got-1, eat-4, ceftriaxone drug, exogenous glutamate treatment) to alter glutamate levels to support the conclusion that increased glutamate in ubr-1 mutants contributes to Ivermectin resistance

Weaknesses:

- The primary target of Ivermectin is GluCls so it is not surprising that alteration of GluCl expression or function would lead to Ivermectin resistance

- It remains to be seen what percent of Ivermectin resistant parasites in the wild have disrupted glutamate homeostasis as opposed to mutations that more directly decrease GluCl expression or function.

Comments on revisions: All my concerns have been addressed by the authors.

---

## [Referee Report · Reviewer #2 (Public review)]

Summary:

The authors provide a very thorough investigation on the role of UBR-1 in anthelmintic resistance using the non-parasitic nematode, *C. elegans*. Anthelmintic resistance to macrocyclic lactones is a major problem in veterinary medicine and likely just a matter of time until resistance emerges in human parasites too. Therefore, this study providing novel insight into the mechanisms of ivermectin resistance is particularly important and significant.

Strengths:

The authors use very diverse technologies (behavior, genetics, pharmacology, genetically encoded reporters) to dissect the role of UBR-1 in ivermectin resistance. Deploying such a comprehensive suite of tools and approaches provides exceptional insight into the mechanism of how UBR-1 functions in terms of ivermectin resistance.

Weaknesses:

I do not see any major weaknesses in this study. My only concern is whether the observations made by the authors would translate to any of the important parasitic helminths in which resistance has naturally emerged in the field. This is always a concern when leveraging a non-parasitic nematode to shed light on a potential mechanism of resistance of parasitic nematodes, and I understand that it is likely beyond the scope of this paper to test some of their results in parasitic nematodes.

Comments on revisions: The authors have now addressed all my concerns.

---

## [Referee Report · Reviewer #3 (Public review)]

Summary:

Li et al propose to better understand the mechanisms of drug resistance in nematode parasites by studying mutants of the model roundworm *C. elegans* that are resistant to the deworming drug ivermectin. They provide compelling evidence that loss-of-function mutations in the E3 ubiquitin ligase encoded by the UBR-1 gene make worms resistant to the effects of ivermectin (and related compounds) on viability, body size, pharyngeal pumping rate, and locomotion and that these mutant phenotypes are rescued by a UBR-1 transgene. They propose that the mechanism is resistance is indirect, via the effects of UBR-1 on glutamate production. They show mutations (vesicular glutamate transporter eat-4, glutamate synthase got-1) and drugs (glutamate, glutamate uptake enhancer ceftriaxone) affecting glutamate metabolism/transport modulate sensitivity to ivermectin in wild type and ubr-1 mutants. The data are generally consistent with greater glutamate tone equating to ivermectin resistance. Finally, they show that manipulations that are expected to increase glutamate tone appear to reduce expression of the targets of ivermectin, the glutamate-gated chloride channels, which is known to increase resistance.

There is a need for genetic markers of ivermectin resistance in livestock parasites that can be used to better track resistance and to tailor drug treatment. The discovery of UBR-1 as a resistance gene in *C. elegans* will provide a candidate marker that can be followed up in parasites. The data suggest Ceftriaxone would be a candidate compound to reverse resistance.

Strengths:

The strength of the study is the thoroughness of the analysis and the quality of the data. There can be little doubt that ubr-1 mutations do indeed confer ivermectin resistance. The use of both rescue constructs and RNAi to validate mutant phenotypes is notable. Further, the variety of manipulations they use to affect glutamate metabolism/transport makes a compelling argument for some kind of role for glutamate in resistance.

Weaknesses:

The use of single ivermectin dose assays can be misleading. A response change at a single dose shows that the dose-response curve has shifted, but the response is not linear with dose, so the degree of that shift may be difficult to discern and may result from a change in slope but not EC50.

---

## [Author Response]

The following is the authors’ response to the original reviews.

**Public Reviews:**

**Reviewer #1 (Public review):**
Summary:The drug Ivermectin is used to effectively treat a variety of worm parasites in the world, however resistance to Ivermectin poses a rising challenge for this treatment strategy. In this study, the authors found that loss of the E3 ubiquitin ligase UBR-1 in the worm *C. elegans* results in resistance to Ivermectin. In particular, the authors found that ubr-1 mutants are resistant to the effects of Ivermectin on worm viability, body size, pharyngeal pumping, and locomotion. The authors previously showed that loss of UBR-1 disrupts homeostasis of the amino acid and neurotransmitter glutamate resulting in increased levels of glutamate in C. elegans. Here, the authors found that the sensitivity of ubr-1 mutants to Ivermectin can be restored if glutamate levels are reduced using a variety of different methods. Conversely, treating worms with exogenous glutamate to increase glutamate levels also results in resistance to Ivermectin supporting the idea that increased glutamate promotes resistance to Ivermectin. The authors found that the primary known targets of Ivermectin, glutamate-gated chloride channels (GluCls), are downregulated in ubr-1 mutants providing a plausible mechanism for why ubr-1 mutants are resistant to Ivermectin. Although it is clear that loss of GluCls can lead to resistance to Ivermectin, this study suggests that one potential mechanism to decrease GluCl expression is via disruption of glutamate homeostasis that leads to increased glutamate. This study suggests that if parasitic worms become resistant to Ivermectin due to increased glutamate, their sensitivity to Ivermectin could be restored by reducing glutamate levels using drugs such as Ceftriaxone in a combination drug treatment strategy.Strengths:(1) The use of multiple independent assays (i.e., viability, body size, pharyngeal pumping, locomotion, and serotonin-stimulated pharyngeal muscle activity) to monitor the effects of Ivermectin(2) The use of multiple independent approaches (got-1, eat-4, ceftriaxone drug, exogenous glutamate treatment) to alter glutamate levels to support the conclusion that increased glutamate in ubr-1 mutants contributes to Ivermectin resistance.Weaknesses:(1) The primary target of Ivermectin is GluCls so it is not surprising that alteration of GluCl expression or function would lead to Ivermectin resistance.(2) It remains to be seen what percent of Ivermectin-resistant parasites in the wild have disrupted glutamate homeostasis as opposed to mutations that more directly decrease GluCl expression or function.

Thank you for your thoughtful and constructive comments. We completely agree with your observation that alterations in GluCl expression or function can lead to Ivermectin resistance. However, we would like to emphasize that our study highlights an additional mechanism: disruptions in glutamate homeostasis can also lead to decreased GluCl expression, thereby contributing to Ivermectin resistance. This mechanism, which has not been fully explored previously, offers new insights into the complexity of drug resistance and could have important implications for understanding the development of Ivermectin resistance in parasitic nematodes.

As you pointed out, the role of disrupted glutamate homeostasis in wild parasitic populations and the proportion of resistant parasites with this mechanism remain unknown. We believe this uncertainty underlines the significance of our findings, as they suggest a novel avenue for studying Ivermectin resistance and for developing potential strategies to counteract it.

We have incorporated this discussion into the revised manuscript to further enrich the context of our findings.

**Reviewer #2 (Public review):**
Summary:The authors provide a very thorough investigation of the role of UBR-1 in anthelmintic resistance using the non-parasitic nematode, *C. elegans*. Anthelmintic resistance to macrocyclic lactones is a major problem in veterinary medicine and likely just a matter of time until resistance emerges in human parasites too. Therefore, this study providing novel insight into the mechanisms of ivermectin resistance is particularly important and significant.Strengths:The authors use very diverse technologies (behavior, genetics, pharmacology, genetically encoded reporters) to dissect the role of UBR-1 in ivermectin resistance. Deploying such a comprehensive suite of tools and approaches provides exceptional insight into the mechanism of how UBR-1 functions in terms of ivermectin resistance.Weaknesses:I do not see any major weaknesses in this study. My only concern is whether the observations made by the authors would translate to any of the important parasitic helminthes in which resistance has naturally emerged in the field. This is always a concern when leveraging a non-parasitic nematode to shed light on a potential mechanism of resistance of parasitic nematodes, and I understand that it is likely beyond the scope of this paper to test some of their results in parasitic nematodes.

Thank you for your kind words and positive feedback on our work. We greatly appreciate your acknowledgment of the diverse technologies and comprehensive approaches we utilized to uncover the role of UBR-1 in ivermectin resistance.

Your concern about whether our findings in *C. elegans* translate to parasitic helminthes in which ivermectin resistance has naturally emerged is both valid and critical. This is indeed a key question we expect to figure out in future studies. Collaborating with parasitologists to investigate whether naturally occurring mutations in *ubr-1* exist in parasitic and non-parasitic nematodes is a priority for us. We hope that these efforts will lead to meaningful discoveries that have a significant impact on both livestock management and medicine.

**Reviewer #3 (Public review):**
Summary:Li et al propose to better understand the mechanisms of drug resistance in nematode parasites by studying mutants of the model roundworm *C. elegans* that are resistant to the deworming drug ivermectin. They provide compelling evidence that loss-of-function mutations in the E3 ubiquitin ligase encoded by the UBR-1 gene make worms resistant to the effects of ivermectin (and related compounds) on viability, body size, pharyngeal pumping rate, and locomotion and that these mutant phenotypes are rescued by a UBR-1 transgene. They propose that the mechanism is resistance is indirect, via the effects of UBR-1 on glutamate production. They show mutations (vesicular glutamate transporter eat-4, glutamate synthase got-1) and drugs (glutamate, glutamate uptake enhancer ceftriaxone) affecting glutamate metabolism/transport modulate sensitivity to ivermectin in wild-type and ubr-1 mutants. The data are generally consistent with greater glutamate tone equating to ivermectin resistance. Finally, they show that manipulations that are expected to increase glutamate tone appear to reduce expression of the targets of ivermectin, the glutamate-gated chloride channels, which is known to increase resistance.There is a need for genetic markers of ivermectin resistance in livestock parasites that can be used to better track resistance and to tailor drug treatment. The discovery of UBR-1 as a resistance gene in *C. elegans* will provide a candidate marker that can be followed up in parasites. The data suggest Ceftriaxone would be a candidate compound to reverse resistance.Strengths:The strength of the study is the thoroughness of the analysis and the quality of the data. There can be little doubt that ubr-1 mutations do indeed confer ivermectin resistance. The use of both rescue constructs and RNAi to validate mutant phenotypes is notable. Further, the variety of manipulations they use to affect glutamate metabolism/transport makes a compelling argument for some kind of role for glutamate in resistance.Weaknesses:The proposed mechanism of ubr-1 resistance i.e.: UBR-1 E3 ligase regulates glutamate tone which regulates ivermectin receptor expression, is broadly consistent with the data but somewhat difficult to reconcile with the specific functions of the genes regulating glutamatergic tone. Ceftriaxone and eat-4 mutants reduce extracellular/synaptic glutamate concentrations by sequestering available glutamate in neurons, suggesting that it is extracellular glutamate that is important. But then why does rescuing ubr-1 specifically in the pharyngeal muscle have such a strong effect on ivermectin sensitivity? Is glutamate leaking out of the pharyngeal muscle into the extracellular space/synapse? Is it possible that UBR-1 acts directly on the avr-15 subunit, both of which are expressed in the muscle, perhaps as part of a glutamate sensing/homeostasis mechanism?

Thank you for your insightful feedback and thought-provoking questions. These are excellent points that have prompted us to critically reconsider our findings and the proposed mechanism.

Several potential explanations could be considered, although we currently lack direct evidence to support this hypothesis: (1) The pharynx likely plays a dominant role in ivermectin resistance, as previously reported (Dent et al., 1997; Dent et al., 2000), and overexpression of UBR-1 in the pharyngeal muscle may exhibit a strong effect on ivermectin sensitivity. (2) It is also possible that pharyngeal muscle cells have the capacity to release glutamate into the extracellular space, which could contribute to the observed effect. (3) Alternatively, UBR-1 expression in the pharyngeal muscle may regulate other indirect pathways affecting extracellular or synaptic glutamate concentrations.

We also appreciate your suggestion that UBR-1 may act directly on AVR-15 in the pharynx. While this is an interesting possibility, UBR-1 is an E3 ubiquitin ligase, and if AVR-15 were a direct target, we would expect UBR-1 to ubiquitinate AVR-15 and promote its degradation. In this case, loss of UBR-1 should inhibit AVR-15 ubiquitination, reduce its degradation, and lead to increased AVR-15 protein levels in the pharynx. However, our experimental data show a reduction, rather than an increase, in AVR-15::GFP levels in *ubr-1* mutants (Figure 4A). This observation suggests that AVR-15 is less likely to be a direct target of UBR-1. To definitively address this hypothesis, a direct assessment of AVR-15 ubiquitination levels in wild-type and *ubr-1* mutant backgrounds would be needed. We agree that this is an important avenue for future investigation.

The use of single ivermectin dose assays can be misleading. A response change at a single dose shows that the dose-response curve has shifted, but the response is not linear with dose, so the degree of that shift may be difficult to discern and may result from a change in slope but not EC50. Similarly, in Figure 3C, the reader is meant to understand that eat-4 mutant is epistatic to ubr-1 because the double mutant has a wild-type response to ivermectin. But eat-4 alone is more sensitive, so (eyeballing it and interpolating) the shift in EC50 caused by the ubr-1 mutant in a wild type background appears to be the same as in an eat-4 background, so arguably you are seeing an additive effect, not epistasis. For the above reasons, it would be desirable to have results for rescuing constructs in a wild-type background, in addition to the mutant background.

Thank you for your detailed feedback and observations.

The potential additive effect you noted in Figure 3C appears to be specific to the body length analysis. In our other three ivermectin resistance assays (viability, pumping rate, and locomotion velocity), this additive effect was not observed. A possible explanation for this is that *eat-4* and *got-1* single mutants inherently exhibit reduced body length compared to wild-type worms (Mörck and Pilon 2006; Greer *et al.* 2008; Chitturi *et al.* 2018), which may give the appearance of an additive effect in this particular assay.

Regarding the use of rescuing constructs, we performed these experiments in the *ubr-1;got-1* and *ubr-1;eat-4* double mutant backgrounds. This was designed to test whether the suppression of *ubr-1*-mediated ivermectin resistance by *got-1* or *eat-4* mutations is indeed due to the functional activity of GOT-1 and EAT-4, respectively. The choice of this setup was to ensure that the double mutant phenotype was fully addressed. In contrast, rescuing constructs of GOT-1 and/or EAT-4 in a wild-type background might not sufficiently reveal the relationship between GOT-1, EAT-4, and UBR-1. However, we are open to further testing your suggestion in the future.

To aid in the interpretation and clarify the apparent effects, we have revised Figure 3 annotation to clearly represent the data and the comparisons being made. We hope this adjustment makes the results more straightforward and easier for readers to understand.

The added value of the pumping data in Figure 5 (using calcium imaging) over the pump counts (from video) in Figure 1G, Figure 2E, F, K, & Figure 3D, H is not clearly explained. It may have to do with the use of "dissected" pharynxes, the nature/advantage of which is not sufficiently documented in the Methods/Results.

Thank you for pointing this out. The behavioral pumping data in Figure 1G, Figure 2E, F, K, & Figure 3D and calcium imaging data in Figure 5 were obtained under different experimental conditions. Specifically, the behavioral assays (pumping rate) were conducted on standard culture plates with freely moving worms, whereas the calcium imaging experiments were performed in a liquid environment with immobilized worms. In the calcium imaging setup, the dissection refers to gently puncturing the epidermis behind head of the worm with a glass electrode to relieve internal pressure, which aids in stabilizing the calcium imaging process and ensures better visualization of pharyngeal muscle activity.

We compared the pharyngeal muscle activity of worms that were not subjected to puncturing the epidermis and found no significant difference when activated by 20 mM serotonin. Therefore, we speculate that there is no direct interaction between the bath solution and the pharynx or head neurons. To avoid confusion, we have removed the term "dissected" from the manuscript and added additional experimental details in the Methods section.

**Recommendations for the authors:**

**Reviewer #1 (Recommendations for the authors):**
(1) The authors propose that ubr-1 mutants are resistant to ivermectin due to persistent elevation of glutamate that leads to a compensatory reduction in GluCl levels and thus resistance to Ivermectin. This model would be strengthened by experiments more directly connecting glutamate, GluCls and Ivermectin sensitivity. For example, does overexpression of a relevant GluCl such as AVR-15 restore Ivermectin sensitivity to ubr-1 mutants? Does Ceftriaxone treatment affect the Ivermectin resistance of worms lacking the relevant GluCls (i.e., avr-15, avr-14 and glc-1)? - The model suggests that Ceftriaxone treatment would have no effect in the latter case.

Thank you for your valuable suggestion. Based on your recommendation, we have performed two additional experiments to strengthen our model. First, we conducted an overexpression experiment of AVR-15 and found that it significantly, though partially, restored ivermectin sensitivity in *ubr-1* mutants (p < 0.01, Supplemental Figure S5D). Second, we tested the effect of Ceftriaxone treatment on the IVM resistance of *avr-15; avr-14; glc-1* triple mutants, which encode the most critical glutamate receptors involved in IVM sensitivity. As expected, we found that Ceftriaxone treatment did not alter the IVM resistance in these triple mutants (Supplemental Figure S5E), supporting the idea that these specific GluCls are key to the observed resistance.

These two experiments provide further support for our proposed model. We have integrated the results into the manuscript, updating the Results section and Supplemental Figure S5D, E, as well as the corresponding Figure Legends.

(2) Line 211 - Ceftriaxone is known to upregulate EAAT2 expression in mammals. Do the authors know if the drug also increases EAAT expression in *C. elegans*?

Thank you for raising this point. To our knowledge, this is the first study to demonstrate the antagonistic effect of ceftriaxone on ivermectin resistance in *C. elegans*, particularly in the context of *ubr-1*-mediated resistance. Ceftriaxone enhances glutamate uptake by increasing the expression of excitatory amino acid transporter-2 (EAAT2) in mammals (Rothstein et al., 2005, Lee et al., 2008). *C. elegans* has six glutamate transporters encoded by *glt-1* and *glt-3–7* (Mano *et al.* 2007).

Compared to testing whether ceftriaxone increases the expression of these EAATs in *C. elegans*, identifying which specific *glt* gene targeted by ceftriaxone may better reveal its mechanism of action. To investigate this, we performed a genetic analysis. In the *ubr-1* mutant, we individually deleted the six *glt* genes and found that ceftriaxone’s ability to restore ivermectin sensitivity was specifically suppressed in the *ubr-1; glt-1* and *ubr-1; glt-5* double mutants (Author response image 1A). This suggests that *glt-1* and *glt-5* may be the targets of ceftriaxone in *C. elegans*. In contrast, ivermectin sensitivity was unaffected in the individual *glt* mutants (Author response image 1B), indicating that a single *glt* deletion may not be sufficient to alter glutamate level or induce GluRs downregulation. Further studies are needed to determine whether ceftriaxone directly increases GLT-1 and GLT-5 expression in *C. elegans* and to explore the underlying mechanisms.

**Author response image 1. sa4fig1:** Glutamate transporter removal inhibits ceftriaxone-mediated restoration of ivermectin sensitivity in *ubr-1*. (**A**) Compared to the *ubr-1* mutants, the *ubr-1; glt-1* and *ubr-1; glt-5* double mutants show enhanced ivermectin resistance under ceftriaxone treatment. (**B**) The *glt* mutants do not show resistance to ivermectin. *****p* < 0.0001; one-way ANOVA test.

(3) Line 64 - as part of the rationale for the study, the authors state that "...increasing reports of unknown causes of IVM resistance continue to emerge...suggesting that additional unknown mechanisms are awaiting investigation." While this may be true, the ultimate conclusion from this study is that decreasing expression of Ivermectin-targeted GluCls causes Ivermectin resistance, which is a known mechanism. The field already knows that Ivermectin targets GluCls and thus decreasing GluCl expression or function would lead to Ivermectin resistance. The authors may want to edit the sentence mentioned above for clarity.

Thanks for the suggestion. We have revised the sentence for clarity: “…, suggesting that previously unrecognized or additional mechanisms regulating GluCls expression may await further investigation.” This revision better reflects the focus on GluCl regulation and clarifies the potential for additional mechanisms to be explored.

(4) The introduction to the serotonin-stimulated pharyngeal Calcium imaging section is a little confusing. The role of the various GluCls in pharyngeal pumping should be defined/clarified in the introduction to the last section (lines 337-341).

Thanks. We have revised and clarified the introduction as follows: “GluCls downregulation was functionally validated by the diminished IVM-mediated inhibition of serotonin-activated pharyngeal Ca2+ activity observed in *ubr-1* mutants. ”

Additionally, the role of the various GluCls in pharyngeal pumping has been clarified:

“Using translational reporters, we found that IVM resistance in *ubr-1* mutants is caused by the functional downregulation of IVM-targeted GluCls, including AVR-15, AVR-14, and GLC-1. These receptors are activated by glutamate to facilitate chloride ion influx into pharyngeal muscle cells, resulting in the inhibition of muscle contractions and the suppression of food intake in *C. elegans*. ”

We hope these revisions address the concerns raised and improve the clarity of this section.

(5) The color code key on the right-hand side of the Raster Plots in Figure 1H should be made larger for clarity.

Revised.

(6) In Figure S3, a legend should be included to define the black and blue box plots.

Thank you for your comment. We have added the following clarification to the figure legend: “Black plots: wild-type, blue plots: *ubr-1* mutants.” This should now make the distinction between the two groups clear.

(7) Figure S4, the brackets above the graphs are misleading. It is not clear which comparisons are being made.

Thank you for your feedback. We have clarified the figure by updating the legend to include the statement: “All statistical analyses were performed against the *ubr-1* mutant.” This clarification is now also included in Figure 3F-I to ensure consistency and avoid any confusion regarding the comparisons being made.

**Reviewer #2 (Recommendations for the authors):**
(1) In Figure 1A: the "trails" table needs more clarification to orient the reader.

To improve clarity and better orient the reader, we have updated Figure 1A by explicitly adding the number of trials and including a statistical analysis of the viability of wild-type and *ubr-1* mutants under different ML conditions. In Figure 1A legend, we have added “we used shades of red to represent worm viability on each experimental plate (n = 50 animals per plate), with darker shades indicating lower survival rates. The viability test was repeated at least 5 times (5 trials).”. These modifications aim to provide a clearer understanding of the data presentation and its significance.

(2) In Figure S2: it would benefit the reader to include the major human parasitic nematodes in the phylogeny and include a discussion of the conservation.

Thank you for your insightful comment. In Figure S2A, we have included the human parasitic nematodes *Onchocerca volvulus*, *Brugia malayi*, and *Toxocara canis*. Unfortunately, other major human parasitic nematodes, such as *Ascaris lumbricoides* (roundworm), *Ancylostoma duodenale* (hookworm), and *Trichuris trichiura* (whipworm), currently lack reported homologs of the *ubr-1* gene.

To provide some context, *Onchocerca volvulus* is a leading cause of infectious blindness globally, affecting millions of people, while Brugia malayi causes lymphatic filariasis, a significant tropical disease. *Toxocara canis* is a zoonotic parasite responsible for serious human syndromes such as visceral and ocular larval migration. Ivermectin remains a primary treatment for these parasitic infections.

Interestingly, while we have identified relevant sequences in *Onchocerca volvulus*, *Brugia malayi*, and *Toxocara canis*, potential mutations in *ubr-1*-like genes in these parasitic nematodes may lead to ivermectin resistance. Sequence comparison analysis could shed light on the risks of such mutations and their relevance to ivermectin treatment failure, warranting further attention. We have added a discussion of this potential risk in the manuscript.

**Reviewer #3 (Recommendations for the authors):**
Minor corrections/suggestions:(1) The level of resistance in ubr-1 is similar to dyf genes. Should double-check ubr-1 mutant is not dyf.

Thank you for your insightful suggestion. We are also interested in this point and designed the following experiments. We first directly tested the Dyf phenotype of *ubr-1* using standard DIO dye staining (Author response image 2A) and found that *ubr-1* clearly show a "dye filling defective" phenotype (Author response image 2B). This raises an interesting question: Could the IVM resistance observed in *ubr-1* be due to its Dyf defect? To address this, we further performed experiment by using Ceftriaxone to test *ubr-1*’s Dyf phenotype. Ceftriaxone can fully rescue the sensitivity of *ubr-1* to IVM (Figure 2). If IVM resistance observed in *ubr-1* is due to its Dyf defect, we should observe same rescued Dyf defect. After treating *ubr-1* mutants with Ceftriaxone (50 μg/mL) until L4 stage, we again performed DIO dye staining and found that while Ceftriaxone fully rescued IVM resistance in *ubr-1*, it did not rescue the Dyf defect (Author response image 2C). These results suggest that while *ubr-1* has a Dyf defect, it is unlikely the primary cause of the IVM resistance in *ubr-1* mutant.

**Author response image 2. sa4fig2:** *ubr-1* mutant is not *dyf.* (**A**) Depiction of the DIO dye-staining assays. Diagram is adapted from Power *et al.* 2020. (**B**) *ubr-1* mutant exhibits obvious Dyf phenotype. (**C**) Cef treatment (50 μg/mL) does not alter the *ubr-1* Dyf defect phenotype. Scale bar, 20 µm.

(2) 367 "in IVM" superscript.(3) 429 ubr-1 italics.

Thanks, revised.

(4) Methods: Need more info on dissection: if there is direct interaction of bath with pharynx, as suggested by bath solution, then 5HT concentrations are too high. Direct exposure to 20mM 5HT will kill a pharynx. 20uM 5HT?

Thank you for your comment. We have reviewed our experimental records and confirmed that the concentration mentioned in the manuscript is correct. In our experiment, the dissection refers to gently puncturing the epidermis behind head of the worm with a glass electrode to relieve internal pressure, which helps stabilize the calcium imaging process. In fact, there is no direct interaction between the bath solution and the pharynx or head neurons. We have revised the Methods section to clarify this point.

(5) Figure 2: Meaning of "Trials" arrow on grid y-axis is not immediately obvious to me. Would prefer you just label/number individual trials.

Sure, we have labeled the trails accordingly in revised Figure 1, 2, and Figure S1.

(6) Figure 3: Legend should include [IVM]. Meaning of +EAT-4, +GOT-1 should be described in the legend.

Thank you for your suggestion. We have updated the figure legend to include the IVM concentration (5 ng/mL). Additionally, we have clarified the meaning of +EAT-4 and +GOT-1 in the legend with the description: “…whereas the re-expression of GOT-1 (+GOT-1) and EAT-4 (+EAT-4) partially reinstated IVM resistance in the respective double mutants.” This ensures the figure is more informative and accessible to the reader.

(7) 784 signalling pathway should just be pathway.

Thanks, revised.

(8) Line 811 " Both types of motor neurons are innervated by serotonin (5 -HT)." Innervated by serotonergic "neurons"? However, even that is misleading because serotonin is not necessarily synaptic.

Thank you for your comment. We have revised the sentence to: “Both types of motor neurons could be activated by serotonin (5-HT).” This clarification better reflects the role of serotonin in modulating motor neuron activity.

(9) Line 814 puffing or perfusion. Perfusion seems more accurate. Make the figure consistent.

Thanks, revised.

(10) Figure S1 requires an x axis label with better explanation.

Thank you for your feedback. We have revised Figure S1 and added "x-axis" to clarify that it represents the trail number. Additionally, we have updated the figure legend to include the experimental conditions: “The shades of red represent worm viability, with darker shades indicating lower survival rates, based on 100 animals per plate and at least 5 trials.”

(11) Figure S2 C-F needs ivermectin concentration.(12) Line 865 plants -> plates?

Thanks, revised.

(13) Figure S4. 875 "Rescue of IVM sensitivity of the ubr-1 mutant by the UBR-1 genomic fragment." Wrong title? Describes GFP expression and RNAi experiments.

Thank you for pointing out the mistake in the title. We have revised the title to: “Knockdown of UBR-1 induces IVM resistance phenotypes.” Additionally, we have updated the figure description to include details about GFP expression and RNAi experiments. The GFP expression is now described as: “Expression of functional UBR-1::GFP, driven by its endogenous promoter, was observed predominantly in the pharynx, head neurons, and body wall muscles with weaker expression detected in vulval muscles and the intestine.” The RNAi experiments are described as: “Double-stranded RNA (dsRNA) interference was employed to suppress gene expression in specific tissues (Methods).”

References:

Catarina Mörck, Marc Pilon. C. elegans feeding defective mutants have shorter body lengths and increased autophagy. BMC Dev Biol. 2006 Aug 3:6:39. doi: 10.1186/1471-213X-6-39.

Elisabeth R Greer, Carissa L Pérez, Marc R Van Gilst, Brian H Lee, Kaveh Ashrafi, Neural and molecular dissection of a C. elegans sensory circuit that regulates fat and feeding.Cell Metab. 2008 Aug;8(2):118-31. doi: 10.1016/j.cmet.2008.06.005.

Itzhak Mano, Sarah Straud, Monica Driscoll. Caenorhabditis elegans glutamate transporters influence synaptic function and behavior at sites distant from the synapse. J Biol Chem. 2007 Nov 23;282(47):34412-9. doi: 10.1074/jbc.M704134200.

Kade M Power, Jyothi S Akella, Amanda Gu et al., Mutation of NEKL-4/NEK10 and TTLL genes suppress neuronal ciliary degeneration caused by loss of CCPP-1 deglutamylase function.PLoS Genet. 2020.16(10):e1009052.doi: 10.1371/journal.pgen.1009052.